



**Chemically speciated mass size distribution, particle effective density and origin of**
**non-refractory PM₁ measured at a rural background site in Central**
Petra Pokorná[1], Naděžda Zíková[1], Radek Lhotka[1,2], Petr Vodička[1], Saliou Mbengue[3], Adéla
Holubová Šmejkalová[4], Véronique Riffault[5], Jakub Ondráček[1], Jaroslav Schwarz[1], Vladimír
Ždímal[1]
[1]Department of Aerosol Chemistry and Physics, Institute of Chemical Process Fundamentals,
Czech Academy of Sciences, Rozvojová 135/1, 165 02 Prague, Czech Republic
[2]Institute for Environmental Studies, Faculty of Science, Charles University, Benátská 2, 128
01 Prague 2, Czech Republic
[3]Global Change Research Institute, Czech Academy of Sciences, Bělidla 986/4a, 603 00
Brno, Czech Republic
[4]Czech Hydrometeorological Institute, Air Quality Division, Na Šabatce 2050/17, 143 06
Prague, Czech Republic
[5] IMT Lille Douai, Institut Mines-Télécom, Université de Lille, Centre for Energy and
Environment, F-59000 Lille, France
*Correspondence to: Petra Pokorná (pokornap@icpf.cas.cz)*
**Abstract**
The seasonal variability of non-refractory PM₁ (NR-PM₁) was studied at a rural background
site (National Atmospheric Observatory Košetice – NAOK) in the Czech Republic to examine
the impact of atmospheric regional and long-range transport in Central Europe. NR-PM₁
measurements were performed by compact time-of-flight aerosol mass spectrometry (C-ToF-
AMS), and the chemically speciated mass size distributions, effective density, and origin were
discussed. The average PM₁ concentrations, calculated as the sum of the NR-PM₁ (after
collection efficiency corrections – CE corrections of 0.4 and 0.33 in summer and winter,
respectively) and the equivalent black carbon (eBC) concentrations measured by an
aethalometer (AE), were $8.58 \pm 3.70$ μg m⁻³ in summer and $10.08 \pm 8.04$ μg m⁻³ in winter.
Organics dominated during both campaigns (summer/winter: $4.97 \pm 2.92 / 4.55 \pm 4.40$ μg m⁻³),
followed by $SO_4^{2-}$ in summer ($1.68 \pm 0.81 / 1.36 \pm 1.38$ μg m⁻³) and $NO_3^-$ in winter ($0.67 \pm$
$0.38 / 2.03 \pm 1.71$ μg m⁻³). The accumulation mode dominated the average mass size distribution
during both seasons, with larger particles of all species measured in winter (mode diameters:
Org: 334/413 nm, $NO_3^-$: 377/501 nm, $SO_4^{2-}$: 400/547 nm, and $NH_4^+$: 489/515 nm) pointing to
regional and long-range transport. However, since the winter aerosols were less oxidized than
the summer aerosols (comparing fragments f₄₄ and f₄₃), the importance of local sources in the
cold part of the year was not negligible. The average PM₁ particle effective density, defined as
the ratio of the mass to the volume of a particle, corresponded to higher inorganic contents
during both seasons (summer: ∼1.30 g cm⁻³ and winter: ∼1.40 g cm⁻³). However, the effective
densities during episodes of higher mass concentrations calculated based on the particle number
(mobility diameter) and mass size distribution (vacuum aerodynamic diameter) were even
higher, ranging from 1.40 – 1.60 g cm⁻³ in summer and from 1.40 – 1.75 g cm⁻³ in winter.
Although aged continental air masses from the SE were rare in summer (7%), they were





connected with the highest concentrations of all NR-PM$_1$ species, especially $SO_4^{2-}$ and $NH_4^+$. In
winter, slow continental air masses from the SW (44%) were linked to inversion conditions
over Central Europe and were associated with the highest concentrations among all NR-PM$_1$
measurements.

## 1. Introduction


Studies on airborne particulate matter (PM) are needed to better understand its temporal and
spatial variations, atmospheric processing, long-term trends, adverse health and environmental
consequences, and pollution sources (Putaud, et al., 2004; Tørseth et al., 2012; Belis et al.,
2013; EEA 2019). Aerosol particles can be characterized by many different properties such
number concentration, mass concentration, particle size, mass, volume, density, etc. Particle
density is an important physical property of atmospheric particles and is linked to particle
emission sources and atmospheric physical and chemical ageing processes. The effective
density, which is defined as the ratio of the mass of the particle to its apparent volume, assuming
a spherical particle, and can be estimated by comparing the size distributions of the
aerodynamic and mobility diameters, is a quantity reflecting the physiochemical properties of
aerosol particles (e.g., DeCarlo 2004; Pitz et al., 2003, 2008; Hu et al., 2012; Qiao et al., 2018).
Over the last decades, a growing number of scientific studies have investigated the detailed
chemical composition of PM with variable temporal resolutions (1, 12, and 24 hours or higher)
using offline filter analyses (Putaud et al., 2010; Watson and Chow, 2011). Nowadays, online
methods with high temporal resolutions (30 min and less) are available, as aerosol mass
spectrometers (AMSs) are utilized that quantitatively measure chemical composition as well as
the chemically resolved size distributions of submicron non-refractory PM (NR-PM$_1$) (Jayne et
al., 2000; Jimenez et al., 2003). Although measuring the seasonal variability of NR-PM$_1$ is
becoming more common (Bressi et al., 2021), systematic studies considering chemically
speciated mass size distributions are still rare. The available studies have also focused on new
particle formation and growth, temporal variations, and the origin and sources of particles,
including results presented from urban (Drewnick et al., 2004; Dall'Osto et al., 2009; Hersey
et al., 2011; Freutel et al., 2013; Salimi et al., 2015; Kubelová et al., 2015), forestry (Allan et
al., 2006), mid-altitude (Freney et al. 2011) and rural (Poulain et al., 2011; Milic et al., 2017)
background environments.
Measurements obtained at rural background sites that are representative of wider areas are
important for investigating the influence of regional and long-range transport as well as the
long-term trends in PM characteristics. In the Czech Republic, the National Atmospheric
Observatory Košetice (NAOK), officially classified as a Central European rural background
site, is involved in the European Monitoring and Evaluation Programme (EMEP), Aerosol,
Clouds, and Trace Gases Research Infrastructure Network (ACTRIS), and Global Atmosphere
Watch (GAW) network. This site has been characterized in terms of the local PM$_{2.5}$ chemical
composition and seasonal variability (Schwarz et al., 2016), the PM$_1$ isotopic composition
(Vodička et al., 2019) and the PAH$_S$ that are bound to PM$_1$ (Křůmal and Mikuška, 2020).
Studies conducted at NAOK have also characterized the long-term trends of atmospheric
carbonaceous aerosols (Mbengue et al., 2018, 2020) and PM$_{2.5}$ elemental compositions and
sources (Pokorná et al., 2018). The particle number size distribution (PNSD) and influence of
in-cloud and below-cloud scavenging have been investigated with long-term measurements by



Zíková and Ždímal (2013, 2016). However, detailed work focused on the seasonal variability
in PM chemical composition data with high temporal and spatial resolutions is still lacking at
this site.
Therefore, this paper aims to assess NR-PM$_1$ (organics, sulphate, nitrate, ammonium and
chloride) based on the chemically speciated mass size distribution, particle effective density,
and origin during intensive campaigns in summer and winter at NAOK.

**2. Materials and methods**
**2.1 Instrumentation**
Two intensive sampling campaigns were carried out in July 2019 (1.7. – 31.7.) and in January-
February 2020 (16.1. – 10.2.) at NAOK. During the campaigns, several physical and chemical
atmospheric aerosol properties were measured together with complete meteorological data
collected from a professional meteorological station (WMO station 11628).
The size-resolved NR-PM$_1$ chemical composition (the sum of organic, sulphate, nitrate,
ammonium and chloride) was measured by a compact time-of-flight aerosol mass spectrometer
(C-ToF-AMS, Aerodyne, USA, Drewnick et al., 2005) with a 5-min temporal resolution. The
instrument was connected to an inlet consisting of a PM$_{2.5}$ sampling head (16.7 l min$^-$) and a
Nafion dryer (Perma Pure MD-110-24P-4). Isokinetic sub-sampling was used to split the flow
into AMS (0.1-l min) from the main flow. The AMS size, flow, and ionization efficiency (IE)
calibrations were performed in the brute-force single-particle mode (BFSP, Drewnick et al.,
2005, monodisperse 350-nm ammonium nitrate aerosol particles) at the beginning of each
campaign. Additionally, the measurements were performed with a HEPA filter applied to the
inlet to account for zero-value measurements and to adjust the fragmentation table (Allan et al.,
110  2004).

Additionally, 12-h PM$_1$ filter samples were collected by a sequential Leckel LVS-3 (Sven
Leckel Ingenieurbüro, Germany) for subsequent chemical analyses of cations, anions and
monosaccharide anhydrides using ion chromatography (Dionex ICS-5000+ system, Sunnyvale,
CA, USA). More details about the methods can be found in Kozáková et al., 2019.
The particle number concentration (PNC) and particle number size distribution (PNSD) were
measured every 5 min by a mobility particle size spectrometer (MPSS, IFT TROPOS, Germany,
with CPC 3772, TSI USA) in the size range of 10 – 800 nm (a detailed description of the
measurement set-up can be found in Zíková and Ždímal, 2013). The cumulative particle number
concentrations over seven size ranges (10 – 25 nm, 25 – 50 nm, 50 – 80 nm, 80 – 150 nm, 150
– 300 nm, 300 – 800 nm, and 10 – 800 nm) were subsequently calculated from the PNSD.
Additionally, the 1-h PM$_{2.5}$ mass concentrations were measured using a beta-gauge (MP101M,
Environement SA, France).
The concentrations of equivalent black carbon (eBC) were estimated using a 7-wavelength
aethalometer (Model AE33, Magee Scientific, Berkeley, CA, USA) sampling through a PM$_{10}$
sampling head (Leckel GmbH) with a 1-min temporal resolution. Additionally, 4-h PM$_{2.5}$ online
organic and elemental carbon (OC/EC) concentrations (Sunset Laboratory Inc., USA) were
measured following the shortened EUSAAR2 protocol (Cavalli et al., 2010).





**2.2 Data analysis**
The standard data processing procedure of AMS data (i.e., m/z calibration, baseline subtraction,
and air beam correction) was carried out by running the Squirrel v1.62 program in Igor Pro data
analysis software (WaveMetrics, Inc.).
The statistical data treatment was performed using R version 3.6.1 (R Core Team, 2019) with
the ggplot2 (Wickham, 2016) and Openair (Carslaw and Ropkins, 2012) packages.
**2.2.1 Collection efficiency determination**
To determine the collection efficiency (CE; Drewnick et al., 2005) in the AMS, $PM_1$ filter
sampling with subsequent ion chromatography (IC) analysis was conducted in parallel with the
AMS measurements. A comparison between the sulphate concentrations measured by AMS
and by IC revealed the better suitability of the CE corrections for summer (CE = 0.40; y =
0.99x, $R^2$ = 0.95) as well as for winter (CE = 0.33; y = 1.00x, $R^2$ = 0.81) in comparison to the
composition-dependent CE correction (CDCE; Middlebrook et al., 2012) shown in Fig. A1.
Therefore, CE correction was applied to the AMS data for both seasons to maintain consistency
in the data corrections. Similarly, using the same methodology, seasonal CE corrections
(summer CE = 0.29 and winter CE = 0.35) were also successfully applied to AMS data
measured at a suburban site in Prague (Kubelová et al., 2015).
**2.2.2 Particle effective density calculation**
Two approaches were employed to calculate the particle effective density. In the first approach,
AMS data representing the mass size distributions based on the vacuum aerodynamic diameter
($D_{va}$) in the size range from 10 to 7000 nm (calculated in Squirrelu software; 50 – 800 nm in
reality) and MPSS data representing the dN/dlog $D_p$ in the size range from 11.3 to 987 nm were
utilized. In the MPSS data, he $D_{va}$ were recalculated using the mobility diameters with a density
of 1.5 g cm$^{-3}$, and the $D_{va}$ were then recalculated back to mobility diameters with the
assumption of spherical particles as in DeCarlo et al. (2004):
$$D_m = \frac{D_{va}}{\rho}\rho_0, \tag{1}$$

where $D_m$ is the mobility diameter, $D_{va}$ is the vacuum aerodynamic diameter, $\rho_0$ is the water
density, and $\rho$ is the total density of particles, resulting in the sizes ranging from 7.53 to 658
nm. The position of the main mode was compared between the AMS and MPSS data to estimate
the aerosol effective density. The density was first used to recalculate the diameters and was
later also used for the mass calculations. The dN data were calculated and used for the dV and
dM distribution calculations.

In the second approach, the mass concentrations of NR-$PM_1$ species and eBC were converted
to the estimated size-dependent density ($\rho$) based on the following equation from Salcedo et al.
163  (2006).

$$\rho = \frac{[Total_{AMS} + eBC]}{\frac{[NO_3^-] + [SO_4^{2-}] + [NH_4^+]}{1.75} + \frac{[Cl^-]}{1.52} + \frac{[Org]}{1.20} + \frac{[eBC]}{1.77}} \tag{2}$$

The densities were assumed to be approximately 1.75 g cm$^{-3}$ for ammonium nitrate, ammonium
sulphate, and ammonium bisulphate (Lide, 1991); 1.52 g cm$^{-3}$ for ammonium chloride (Lide,



1991); 1.20 g cm$^{-3}$ for organics (Turpin and Lim, 2001); and 1.77 g cm$^{-3}$ for black carbon (Park
et al., 2004).

**2.2.3 Cluster analysis**

For both campaigns, 96-hour backwards trajectories were calculated using the Hybrid Single-
Particle Lagrangian Integrated Trajectory (HYSPLIT) model (Rolph et al., 2017) with a 500-m
AGL starting position and Global Data Assimilation System (GDAS) Archive Information at a
resolution of 1° × 1° as input data. The calculations were initialized every 6 hours. The
trajectories were further clustered using Hysplit4 software based on the total spatial variance.
From HYSPLIT, the planetary boundary layer height data were extracted using the vmixing
program (https://www.ready.noaa.gov/HYSPLIT_vmixing.php). For the planetary boundary
layer height calculations, the 0.25° × 0.25° Global Forecast System (GFS) dataset was used as
input data to obtain a 3-hour temporal resolution.

**2.2.4. Episodes of high particle number and mass concentrations**

To determine episodes of high particle number and mass concentrations, two approaches were
utilized: i) the application of positive matrix factorization (PMF) to PNSDs and ii) the depiction
of the mass size distribution of NR-PM$_1$ species. The episodes were studied in detail from the
particle effective density and mass size distribution perspectives.

**2.2.4.1 PMF on PNSD**

PMF (US EPA, version PMF 5.0) was applied to the seasonal 5-min PNSDs in the range from
10 nm to 800 nm to estimate the number and profile of the PNSD factors and their contributions
to the receptor. Episodes in which the factor contributions to the total particle number
concentrations were higher than 80 % were chosen for the subsequent particle effective density
calculations.
The input data were prepared by merging three consecutive bins to reduce the noise in the raw
data, decrease the number of variables, and reduce the number of zeroes in the raw data (Leoni
et al., 2018). The uncertainties were calculated according to Vu et al. (2015). The total variables
were calculated by summing all the bins (N10 − 800). PMF was conducted using different
uncertainty input matrices and different C3 (Vu et al., 2015) to obtain the Q$_{true}$ closest to
Q$_{expected}$; different modelling uncertainties and different numbers of factors were also applied.
A C3 of 0.8 was chosen.

**2.2.4.2 3D plots**

The mass size distributions of nitrate, sulphate and organic matter are depicted in a colour-
coded 3D plot showing episodes of high mass concentrations.


**3. Results and discussions**

**3.1 Campaign overview**

The campaigns were characterized by prevailing westerly winds with average wind speed of
3.0 ± 1.5 m s$^{-1}$ in summer and 4.5 ± 3.1 m s$^{-1}$ in winter (Fig. A2), average temperature of 18.5



± 4.7 °C in summer and 1.4 ± 3.9 °C in winter, and negligible precipitation. The average PM$_{2.5}$
was 10.9 ± 5.9 µg m$^{-3}$ in summer and 11.8 ± 9.9 µg m$^{-3}$ in winter (2019 average annual
concentration: 10.1 µg m$^{-3}$, CHMI, 2019a).
Based on the PNSD, in summer, particles in the size range of 25 – 80 nm (N25 – 50 and N50 –
80) predominated, while in winter, N80 – 150 were dominant (Table 1). Particles in the size
range of 25 – 80 nm, also called the Aitken mode, are typical for rural background stations and
originate from the ageing of particles generated during new particle formation (NPF) events
(Costabile et al., 2009). Based on a 5-year study (2013 – 2017) evaluating PNSDs at NAOK,
June and July were classified as the months with the highest NPF event frequencies (38 and
36% of days, respectively, Holubová Šmejkalová et al., 2021). The prevailing accumulation-
mode particles in winter were presented in Schwarz et al., 2016, as well as in Zíková and Ždímal
(2013). The average PNCs recorded during the two studied seasons were lower than the annual
mean total concentration (6.6 × 10$^3$ cm$^{-3}$, Zíková and Ždímal, 2013).
Table 1. Average cumulative particle number concentrations (cm$^{-3}$) measured by MPSS during
the summer and winter campaigns.

| Size range (nm) | Summer | Winter |
|---|---|---|
| N10 – 25 | 979±1488 | 315±344 |
| N25 – 50 | 1726±1536 | 529±402 |
| N50 – 80 | 1112±715 | 478±492 |
| N80 – 150 | 907±472 | 606±654 |
| N150 – 300 | 508±191 | 437±368 |
| N300 – 800 | 51±41 | 86±76 |
| N10 – 800 (Total) | 4971±2794 | 2451±1749 |


## 3.2 Volume and mass closure analysis with PNSD

For the mass closure analysis, the total mass concentrations measured by AMS (the sums of the
organic, sulphate, nitrate, ammonium and chloride concentrations) were complemented by the
eBC mass concentrations. The average PM$_1$ concentrations for the summer and winter
campaigns were 8.58±3.70 µg m$^{-3}$ (filter-based 12-hour PM$_1$ 10.10 ± 6.44 µg m$^{-3}$) and 10.08 ±
8.04 µg m$^{-3}$ (filter-based 12-hour PM$_1$ 11.05 ± 7.22 µg m$^{-3}$), respectively. Since the PNSD (10-
to 800-nm mobility diameter) was measured continuously in parallel with the eBC and NR-
PM$_1$ mass, mass closure of the 10-min averages was performed. To do so, two approaches were
utilized: i) converting the NR-PM$_1$ + eBC mass concentrations into volume concentrations
using the composition-dependent density and ii) converting the PNSDs into mass
concentrations using a constant density of 1.5 g cm$^{-3}$. Over the summer campaign, the NR-PM$_1$
+ eBC volume and mass concentrations agreed well with the MPSS volume and mass
concentrations in comparison to the winter campaign (Fig. 1). The seasonal effect on mass
closure—already reported by Poulain et al., 2020 using ACSM at rural Melpitz, as well as by
Fröhlich et al., 2015 using ToF-ACSM at Jungfraujoch could be explained by higher
concentrations in larger size bins of the volume size distribution in winter compared to in
summer (Fig. 2), since the AMS underestimates the particle mass concentrations for the larger
size bins. This is due to the specific size cutting of each instrument and the transmission
efficiency of the aerodynamic lens (Poulain et al., 2020). Moreover, the constant density is a
limitation of the mass approach due to the density variability within the distinct episodes.





Irregularities in the mass size distributions of nitrate, sulphate, and ammonia are discussed
further in this paper.

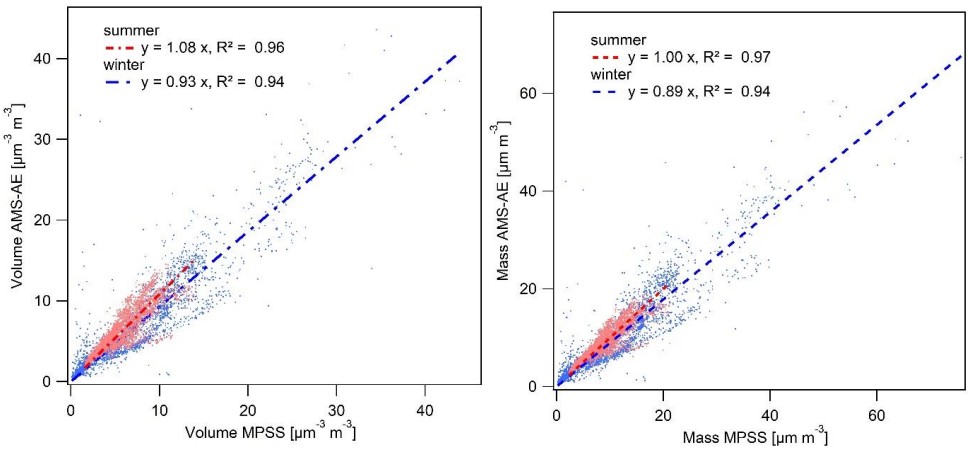


Fig. 1. Comparison between the AMS-AE and MPSS measurements during both campaigns:
volume closure (left) and mass closure (right).

**3.3 Concentration and origin of NR-PM$_1$**

The CE-corrected mass concentrations of NR-PM$_1$ species, calculated as functions of time
during the two campaigns, are shown in Fig. A3. and the seasonal average concentrations are
presented in Table 2. Organics dominated during both campaigns, followed by $SO_4^{2-}$ in summer
and $NO_3^-$ in winter. The PM$_1$ IC results confirmed higher mean $SO_4^{2-}$ concentrations in summer
($SO_{4\,IC}^{2-}$ 1.63 ± 0.84 µg m$^{-3}$ and $NO_{3\,IC}^-$ 0.23 ± 0.18 µg m$^{-3}$). However, the mean $NO_3^-$
concentrations were slightly lower than the $SO_4^{2-}$ concentrations in winter ($NO_{3\,IC}^-$ 0.72 ± 0.52
µg m$^{-3}$ and $SO_{4\,IC}^{2-}$ 0.78 ± 0.58 µg m$^{-3}$). The difference between the $NO_3^-$ concentrations in NR-
PM$_1$ and PM$_1$ for both seasons could be explained by the loss of ammonium nitrate from the
filter due to its dissociation into its gaseous precursors. Good agreement was obtained between
the summer average NR-PM$_1$ $NH_4^+$ and PM$_1$ $NH_4^+$ (0.80 ± 0.37 µg m$^{-3}$ vs 0.70 ± 0.36 µg m$^{-3}$)
in comparison to those obtained in winter (1.11 ± 0.99 µg m$^{-3}$ vs 0.46 ± 0.35 µg m$^{-3}$). The
seasonal variability in nitrate, which displayed higher concentrations in winter, was related to
the thermal instability of ammonium nitrate (Seinfeld and Pandis, 2006). A higher share of Cl$^-$
on NR-PM$_1$ in winter (3 %) indicates the influence of coal combustion used for domestic
heating (CHMI, 2019b).
Overall, the average $SO_4^{2-}$ concentration obtained in this study was lower than that measured
at the Melpitz rural background site (2.44 µg m$^{-3}$ in summer and 1.66 µg m$^{-3}$ in winter, Poulain
et al., 2011) and lower than the values presented in previous studies by Schwarz et al. (2016)
conducted at NAOK (PM$_{2.5}$ IC 2.30 µg m$^{-3}$ in summer and 3.86 µg m$^{-3}$ in winter) and by
Kubelová et al. (2015) conducted in a Prague urban background site (2.0 µg m$^{-3}$ in summer and
4.4 µg m$^{-3}$ in winter). The average summer $NO_3^-$ concentration was comparable to those
measured in Melpitz (0.66 µg m$^{-3}$), NAOK (PM$_{2.5}$ IC 0.55 µg m$^{-3}$) and Prague (0.80 µg m$^{-3}$);
however, the winter average concentration was lower than those reported in all three studies
(Melpitz: 3.62 µg m$^{-3}$, NAOK: 2.83 µg m$^{-3}$, Prague: 5.40 µg m$^{-3}$). The average organic





concentration was lower in summer but higher in winter than the values recorded in Melpitz
(6.89 µg m$^{-3}$ (51%) and 2.08 µg m$^{-3}$ (23%), respectively). The comparison of organic mass
(OM) by AMS and OC using an OCEC field analyser is shown in Fig. A4. Turpin and Lim,
2001 recommended OM/OC ratio of 2.1 for non-urban (aged) particles and of 1.6 for urban
particles. In this study, the average OM/OC ratio was 2.06 (±0.68) in summer and 1.51 (±0.36)
in winter. An average OM$_1$ and OC$_{2.5}$ of 2.1±1.4 was determined at the Hohenpeissenberg rural
site in spring, referring to continental OA (Hock et al., 2002). The higher summer OM/OC ratio
could be explained by the presence of more oxidized organic compounds, as the products of
photochemical reactions increase the average organic molecular weight per carbon weight
(Turpin and Lim, 2001). This result is consistent with the increasing OC/EC ratio observed
during summer, when photochemical activity leads to larger secondary organic carbon
formation (Mbengue et al., 2018, 2020). Another explanation could be the increased boundary
layer height, which enables mixing from higher altitudes and therefore the entrainment of aged,
and thus more oxidized, aerosols from long-range transport (Querol et al., 1998). On the other
hand, the winter season is characterized by fresh emissions of hydrocarbons owing to the
lowered boundary layer height in winter, which does not support the transport of oxidized
pollutants within the mixing layer (Schwarz et al., 2008).
Table 2. Basic statistics of the NR-PM$_1$ and eBC concentrations (median, mean, standard
deviation (SD) and average share of species in the total concentration) measured during summer
and winter. The values were calculated from five-min-resolution CE-corrected data.

| Summer | Org | $SO_4^{2-}$ | $NO_3^-$ | $NH_4^+$ | $Cl^-$ | eBC |
|---|---|---|---|---|---|---|
| Median (µg m$^{-3}$) | 4.32 | 1.53 | 0.57 | 0.75 | 0.06 | 0.36 |
| Mean (µg m$^{-3}$) | 4.97 | 1.68 | 0.67 | 0.80 | 0.06 | 0.40 |
| SD | 2.92 | 0.81 | 0.38 | 0.37 | 0.02 | 0.20 |
| Average share on NR-PM$_1$ | 58 % | 22 % | 9 % | 10 % | 1 % | -- |
| **Winter** | | | | | | |
| Median (µg m$^{-3}$) | 3.35 | 0.98 | 1.67 | 0.93 | 0.16 | 0.84 |
| Mean (µg m$^{-3}$) | 4.55 | 1.36 | 2.03 | 1.11 | 0.18 | 0.92 |
| SD | 4.40 | 1.38 | 1.71 | 0.99 | 0.09 | 0.77 |
| Average share on NR-PM$_1$ | 50 % | 14 % | 22 % | 11 % | 3 % | -- |


Fig. 2. shows the variations in the particle number and volume and in the sulphate, nitrate and
organic size distributions as function of time. In summer, several NPF episodes were recorded
(Zíková and Ždímal, 2013; Holubová Šmejkalová et al., 2021); however, accumulation-mode
particles were prominent in volume and species mass size distributions. The accumulation mode
of $SO_4^{2-}$ does not show a large amount of variation, indicating a regional origin. In contrast,
$NO_3^-$ shows dial variations in mass concentrations corresponding to the local photochemical
formation of this species (Fig. A5). In winter, the accumulation mode dominated all
distributions and was linked to regional and/or long-range transport (see 3.4 Size distribution
of NR-PM$_1$).

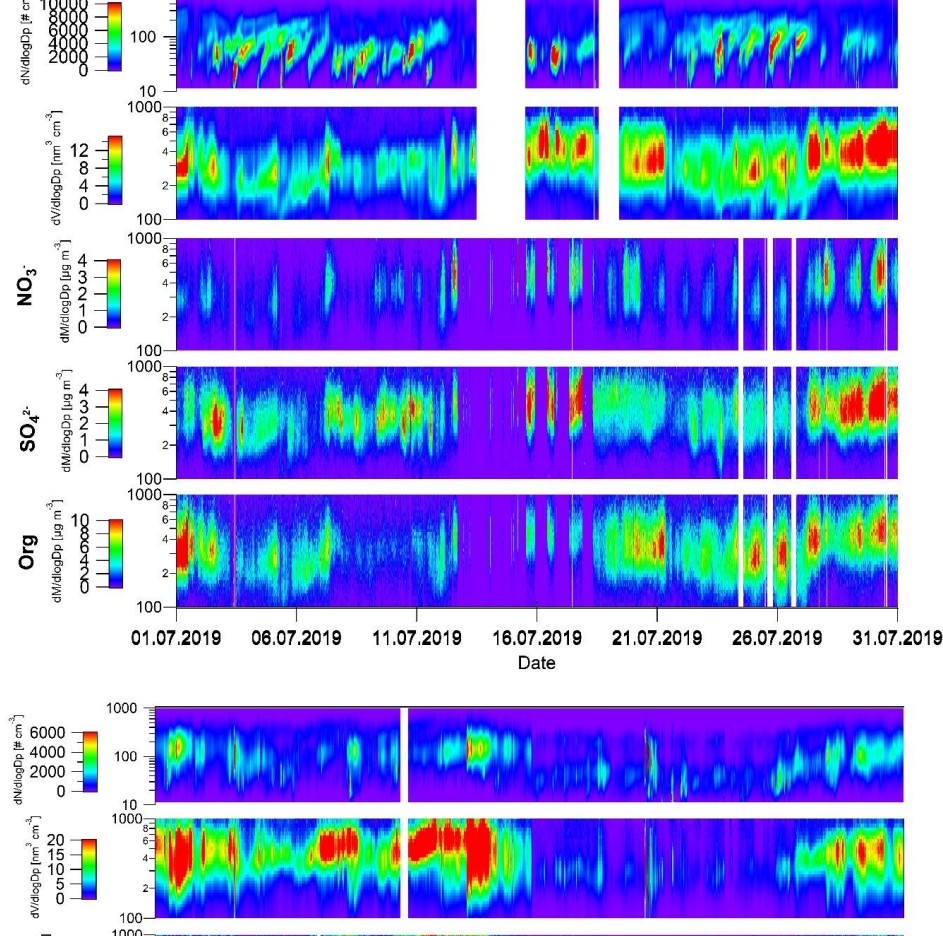


Fig. 2. Time series of particle number and volume concentrations obtained by MPSS ($D_{va}$ recalculated from mobility diameter using a density of 1.50 g cm$^{-3}$) and mass size distributions of nitrate, sulphate and organics obtained by AMS in summer (top) and in winter (bottom).





Based on the mass size distributions of the species (Fig. 2), ten summer (S1 – 10) and 13 winter
(W1 – 13) high-concentration episodes were selected (Table A1). The organic mass dominated
in summer; however, distinct episodes of high $SO_4^{2-}$ concentrations (S2, S8, S9, S10) linked to
continental air masses from the NW and S-SE were also recorded (Fig. A6). In winter, episodes
of dominant $SO_4^{2-}$ (W10) and $NO_3^-$ (W1, W2, W4, W5, W6) concentrations were observed.
W10 was influenced by fresh marine air masses reaching NAOK over the UK, Benelux and
Germany. The episodes of high $NO_3^-$ concentrations were linked to fresh marine air masses
(from the NW) as well as continental air masses (from the NW-SW, Fig. A7).
In summer, the highest Org concentrations (14.58 µg m$^{-3}$) together with the lowest $SO_4^{2-}$ and
$NH_4^+$ (1.24 µg m$^{-3}$ and 0.91 µg m$^{-3}$) concentrations were observed during the S1 night-morning
episode linked to western continental air masses (Table A1 and Fig. A3). S10 represents the
night-morning-early afternoon episode of the highest concentrations of $SO_4^{2-}$, $NO_3^-$ and $NH_4^+$
(6.14 µg m$^{-3}$, 3.37 µg m$^{-3}$, and 2.98 µg m$^{-3}$, respectively) resulting from mixed continental air
masses (NW-S) that were potentially influenced by emissions from coal power plants situated
in North Bohemia.

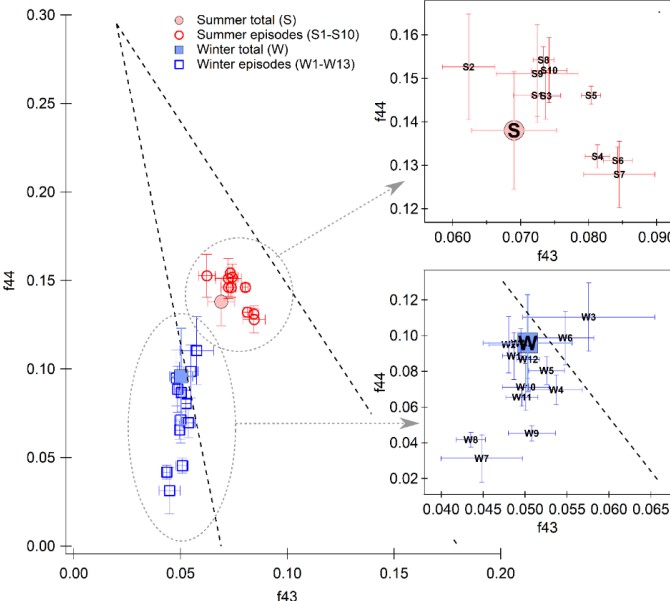


Fig. 3. Comparison of organic fragments f$_{44}$ and f$_{43}$ for the whole summer and winter campaigns
(full markers) and for specific episodes (empty markers). Bars represent the standard deviation
and the triangular space area typical for ambient OOAs (Ng et al., 2010).
The highest concentrations of Org (15.63 µg m$^{-3}$) as well as low concentrations of $SO_4^{2-}$, $NO_3^-$
and $NH_4^+$ (0.74 µg m$^{-3}$, 0.93 µg m$^{-3}$ and 0.96 µg m$^{-3}$, respectively) measured in winter during
W7 were influenced by maritime air masses (Fig. A7). Nevertheless, a one-day inversion
preceded this episode (Fig. A3), characterized by less oxidized OA (Fig. 3) and f$_{44}$f$_{60}$ trending
towards f$_{60}$ (Fig. A8). In contrast, the highest winter $SO_4^{2-}$ and $NH_4^+$ concentrations (7.13 µg m$^{-3}$
and 7.90 µg m$^{-3}$, respectively) measured in the W3 episode and the highest $NO_3^-$





concentrations (10.66 μg m$^{-3}$) measured in the W6a episode were characterized by below-freezing temperatures, which probably arose due to inversion conditions in Central Europe.

Organic aerosol ageing was examined on the f$_{44}$ and f$_{43}$ fragments (Fig. 3). Winter aerosols were less oxidized than summer aerosols, pointing to the importance of local sources during the cold part of the year. In summer, the oxidation rate of organic aerosols within the episodes does not differ greatly, and most of the episodes revealed more oxidized organic aerosols (MOOAs) or less volatile organic aerosols (LV-OOAs). Within the summer campaign, the most oxidized aerosols were detected during afternoon episode S2 (Fig. 3), at which time the highest global radiation was also measured (Table A1.). In contrast, S4, S6 and S7 represent night-time and early morning episodes, and S5 represents a night-time and morning episode, and thus less oxidized aerosols (Fig. 3). In winter, the difference between the episodes is more obvious, mainly due to the higher variability in the local sources that influence the receptor site. The W7, W8 and W9 (Fig. 3) episodes are exceptions; these episodes were linked to fresh marine air masses (Fig. A7.).

The f$_{60}$ fragment was used as a biomass-burning (BB) marker. If ambient aerosols are characterized by f$_{60}$ higher than 0.003, they are considered to be influenced by BB emissions (Cubison et al., 2011). During both campaigns, the average f$_{60}$ was 0.003, in contrast to the presence of levoglucosan in the PM$_1$ samples during both seasons (summer average $0.02 \pm 0.02$ μg m$^{-3}$ and winter average $0.18 \pm 0.20$ μg m$^{-3}$). Levoglucosan concentrations point to BB influence, which was similarly discussed in previous studies conducted at NAOK by Schwarz et al. (2016) and Mbengue et al. (2020). Additionally, a comparison of fragments f$_{44}$ and f$_{60}$ enabled us to assess the presence of fresh organic aerosols emitted by BB (e.g., Milic et al., 2017) revealing that aged organic aerosols from BB influenced the site during both seasons (Fig. A9). The comparison of organic fragments f$_{44}$ and f$_{60}$ determined at the rural and urban background sites shows a difference in the ageing of BB emissions, with the presence of fresh organic aerosols at the urban site and aged organic aerosols at the rural site in winter (Fig. A9).

To determine the origin of NR-PM$_1$ species, back-trajectories describing their air mass origins were clustered using the HYSPLIT model into 6 and 5 clusters in summer and winter, respectively (Fig. 4.) and linked to the organic, nitrate, sulphate, ammonium and chloride concentrations. A seasonal difference was observed in the air mass back-trajectories, with continental air masses prevailing in summer and marine air masses prevailing in winter.

In summer, cluster #1 (continental air masses from the W-NW, 29%) and cluster #3 (fresh marine air masses from the NW, 28%) were most frequent. Although aged continental air masses from the SE probably related to stable anticyclonic conditions (cluster #6) were rare (7%), they were connected with the highest concentrations of all NR-PM$_1$ species, especially $SO_4^{2-}$, $NH_4^+$ and Cl$^-$. $NO_3^-$ was linked to fresh marine air masses (cluster #4, 7%), and Org was linked to continental air masses coming from the W-NW (#1 and #5, 29% and 19%, respectively) (Fig. 4.).

In winter, slow continental air masses from the SW cluster #1 (44%) prevailed. The air masses remaining over Central Europe, likely under inversion conditions, were associated with the highest concentrations of all NR-PM$_1$ species except Cl$^-$ since there was no statistically significant difference among the clusters at the 0.05 level (Fig. 4.). The high pollution loads over Central Europe agree well with the high average mass concentrations of secondary species



during periods when air masses are advected from Central Europe to Paris (Freney et al., 2011,
Crippa et al., 2013; Freutel et al., 2013, Freney et al., 2014).

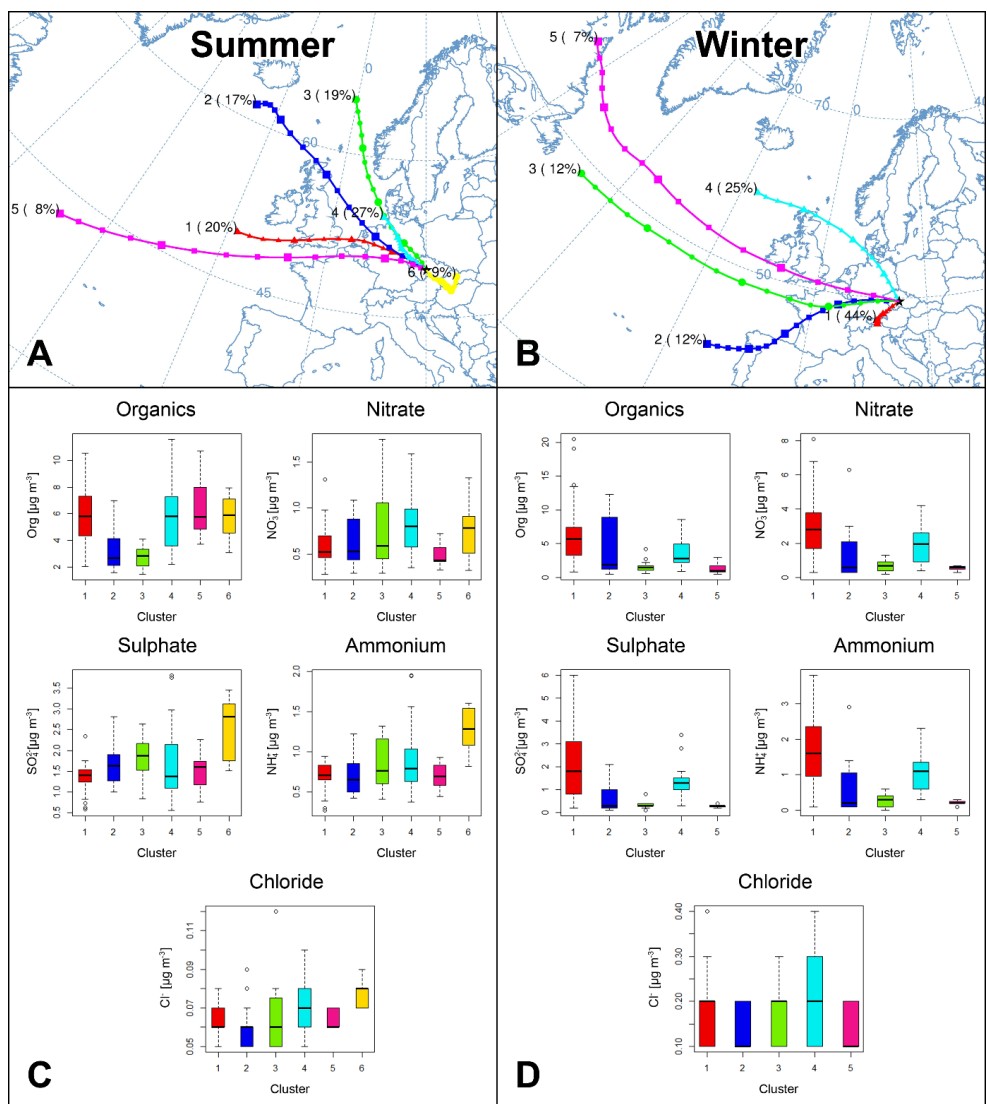


Fig. 4. Geographical locations of the means of the clusters observed in summer (A) and winter
(B) along with boxplots of the organic, nitrate, sulphate, ammonium and chloride
concentrations in individual clusters measured during the summer (C) and winter (D)
campaigns. The boxes are colour coded as the clusters, the black horizontal line is the median,
the boxes border the 25[th] and 75[th] percentiles and the whiskers represent 1.5 x IQR.




### 3.4 Size distribution of NR-PM$_1$

The average mass size distributions of the main NR-PM$_1$ species (except chloride) during the
entire summer and winter campaign are presented in Table 3. To determine the mode diameters
and the widths of the size distributions, the mass distributions were fitted with log-normal
modes using the Igor MultiPeak Package as follows:

$$ y = M\, exp\left[-\left(\frac{\ln(x/x_0)}{width}\right)^2\right], \qquad (3) $$

where $M$ is the amplitude, $x_0$ is the peak position in nm, and *width* denotes the peak width. For
each season, the mean spectra were fitted separately with one peak, and fitting was also
performed for episodes S1-10 and W1-13. Due to the long duration of episode W6, the episode
was split into two sections: W6a (67 hours) and W6b (25.5 hours).
The accumulation mode dominated the average mass size distributions during both campaigns,
with larger particles of all species observed in winter (Table 3). Shifts towards larger $SO_4^{2-}$,
$NO_3^-$ and $NH_4^+$ particles in winter compared to summer were also observed in a previous study
by Schwarz et al. (2012) that determined urban aerosol chemical compositions and size
distributions using a 7-stage impactor with an upstream diffusional aerosol drier. The $SO_4^{2-}$
particles were significantly larger than the $NO_3^-$ particles during both measurement campaigns
except for during two episodes (W7 and W9) with regional transport (Table 3). An accumulation
mode of $SO_4^{2-}$ with regional origin was even detected during a Mexico City Metropolitan Area
field study by Salcedo et al. (2006). Dall'Osto et al. (2009) also observed two nitrate particle
types at an urban background site, both of which were internally mixed with sulphate,
ammonium and carbon: the locally produced particles were smaller than 300 nm , and the
regional particles peaked at 600 nm. In a study by Schwarz et al. (2012) at an urban site in
Prague, two types of $SO_4^{2-}$ particles were determined. $SO_4^{2-}$ particles in sea-influenced aerosol
samples showed maxima between 210 and 330 nm (condensation growth) for both seasons, and
$SO_4^{2-}$ particles in continental-influenced samples showed maxima between 500 and 890 nm in
winter and between 330 and 500 in summer (droplet-phase growth). $NO_3^-$ particles with maxima
between 330 nm and ∼500 nm were observed under maritime and continental air masses during
both seasons. Freutel et al., 2013 observed a single mode of NR-PM$_1$ species of approximately
300 nm under marine air masses as well as a shift of the accumulation mode to a larger size
(approximately 400 nm) during a summer campaign in the Paris region due to aerosol particle
ageing of continental air masses from Central Europe. During a summer measurement
campaign in New York, the average mass distributions of $NO_3^-$, $SO_4^{2-}$ and $NH_4^+$ were
monomodal, with mode diameters of 440 nm, 450 nm and 400 nm, respectively, and the average
Org mass distribution was bimodal, with mode diameters of 80 nm and 360 nm (Drewnick et
al., 2004). A study by Freney et al. (2011) conducted during three seasons at the Puy-de-Dôme
research station presented a major accumulation mode of NR-PM$_1$ species peaking at 600 nm,
indicating aged aerosol particles.





Table 3. Average size distributions of species measured by AMS ($D_p$ corresponds to the vacuum
aerodynamic diameter ($D_{va}$)) for the summer (left) and winter (right) campaigns.

|  | Org | $SO_4^{2-}$ | $NO_3^-$ | $NH_4^+$ |
|---|---|---|---|---|
| Summer $D_{va}$ (nm) | 334 | 377 | 401 | 497 |
| Winter $D_{va}$ (nm) | 413 | 501 | 547 | 517 |

In summer, the smallest mode diameters of Org (279 nm) and $NO_3^-$ (253 nm) were observed
during the S7 episode, while for $SO_4^{2-}$ and $NH_4^+$ (325 nm and 335 nm, respectively), they were
influenced by continental air masses of regional origin during the S2 episode (from the N-NE-
E, Fig. A6). In contrast, the largest mode diameters (Org: 466 nm, $NO_3^-$: 491 mm, $SO_4^{2-}$: 494
nm and $NH_4^+$: 478 nm) were recorded during the S10 episode by continental long-range
transport from the W-NW (Fig. A6). The smallest mode diameters of all species (Org: 295 nm,
$NO_3^-$: 240 nm, $SO_4^{2-}$: 242 nm and $NH_4^+$: 365 nm) in winter (W8) were linked to fresh marine
air masses, and the largest winter diameters (Org: 563 nm, $NO_3^-$: 609 nm, $SO_4^{2-}$: 636 nm and
$NH_4^+$: 607 nm, W3) were linked to the regional and long-range transport of air masses of
continental origin and were also probably influenced by inversion conditions (Fig. A7).
Additionally, as expected, the Org particle size showed growth, and the increasing mode
diameter was more significant in the winter season, with the ageing of aerosols resulting in
oxygenated organic aerosols (Fig. 5).

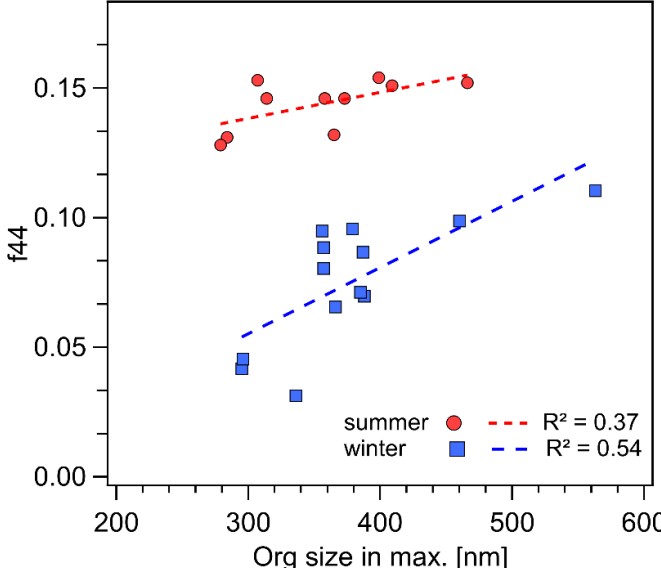

Fig. 5. Relationship between organic fragment $f_{44}$ and the size of the organic fraction during
episodes of high NR-PM$_1$ species mass concentrations in both seasons.







### 3.5 Particle effective density

The particle effective density was calculated for each episode of high particle numbers and mass concentrations. The episodes were determined as follows: i) PMF application to PNSDs and ii) depiction of mass size distributions of NR-PM$_1$ species in a 3D plot (Fig. 2).

The PMF model was run several times until the most physically meaningful results and the best diagnostics were obtained. The two-sided size bins containing variables (9.7 nm, 11.5 nm, 557.2 nm and 733.6 nm; midpoint of the size bins) were set as weak along with the total variables (N10 – 800). The model was run with different factor numbers (3 – 8). The most stable solution was found when 6 factors in summer and 5 factors in winter were considered (Fig. A10). With all runs converged, the scaled residuals were normally distributed, any unmapped factors were detected with bootstrap error estimations. No swaps were observed with the displacement error analysis, indicating that the solution was stable (Table A2). The non-normalized PNSD (N cm$^{-3}$) was analysed using the model.

### 3.5.1 Episodes of high particle number concentrations

One high-particle-contribution episode occurred in summer, and eight short episodes occurred in winter (W1$_{MPSS}$, factor 3 of 5 and W2$_{MPSS}$ – W8$_{MPSS}$, factor 1 of 5; the durations ranged from 25 to 95 minutes). No NR-PM$_1$ data were available for effective density calculations during the summer period (3$^{rd}$ July from 9:20 to 10:05). The effective density ranged between 1.40 and 1.85 g cm$^{-3}$ (Table 4). During W1$_{MPSS}$, accumulation-mode particles dominated (F3, mode diameter ~429 nm, Fig. A10) with an effective density of 1.85 g cm$^{-3}$. A density of 1.85 g cm$^{-3}$ corresponds to black carbon (Martins et al., 1998), and a density of 2.0 g cm$^{-3}$ relates to aged biomass-burning particles (Moffet et al., 2008). The remaining episodes (W2$_{MPSS}$ – W8$_{MPSS}$) were linked mainly to particles of the Aitken mode (F1, mode diameter ~32 nm, Fig. A10) with effective densities ranging from 1.40 to 1.60 g cm$^{-3}$. Rissler et al. (2014) observed the dominance of particles with effective density ~ 1.4 g cm$^{-3}$ at a rural background site (Vavihill, Sweden) during the winter months, and Qiao et al. (2018) reported a decrease in particle effective densities ranging from 1.43 to 1.55 g cm$^{-3}$ at rural sites (Changping, China) with increasing particle sizes.

Table 4. Particle effective densities (g cm$^{-3}$) calculated during episodes of high particle contributions to N10 – 800 using MPSS data.

| Episode MPSS | W1 | W2 | W3 | W4 | W5 | W6 | W7 | W8 |
|---|---|---|---|---|---|---|---|---|
| Density | 1.85 | 1.45 | 1.50 | 1.55 | 1.45 | 1.55 | 1.40 | 1.60 |
| # of spectra | 13 | 8 | 8 | 19 | 7 | 5 | 8 | 8 |

### 3.5.2 Episodes of high mass concentrations

The densities calculated based on the particle mass size distributions using Eq. (1) corresponding to the episodes discussed in section 3.4 (Size distribution of NR-PM$_1$) ranged from 1.40 – 1.60 g cm$^{-3}$ in summer and from 1.30 – 1.75 g cm$^{-3}$ in winter (Table 5, Fig. A11 and Fig. A12). In comparison, the densities calculated using Eq. (2) were lower in both seasons, ranging from 1.30 to 1.40 g cm$^{-3}$ in summer (with a seasonal average of 1.34 ± 0.28 g cm$^{-3}$) and from 1.30 to 1.50 g cm$^{-3}$ in winter (with a seasonal average of 1.44 ± 0.16 g cm$^{-3}$) (Table 5).





The average summer density did not show a diurnal trend compared to the winter density (Fig.
A13), followed by a diurnal trend (inverse dependence) observed for organics (Fig. A5). The
summer diurnal variation in the concentrations of organics was flatter than that in winter and
was not sufficient to significantly affect the diurnal density trend. In summer, we observed the
most significant diurnal trend for nitrate, but the absolute concentrations of nitrate were low,
and this variation therefore did not significantly affect the summer diurnal density trend (Fig.
A5).
In summer, with a higher ratio of ammonium sulphate, the density increased. In winter, the
density was influenced by the inorganic content (ammonium nitrate and sulphate). In both
seasons, the density increased with a decrease in the organic ratio and vice versa. This relation
evidently arises from the parameters in Eq. (2) (Fig. A14). The largest uncertainty in the PM
density calculations performed using Eq. (2) is linked to the density of organics, which was set
to 1.2 g cm$^{-3}$. The density applied for the organic fraction refers to the urban and urban
background stations (Turpin and Lim, 2001), and the organics density of a rural background
site is expected to be higher than that of an urban site due to organic aerosol ageing. However,
a density of 1.2 g cm$^{-3}$ was also utilized in a study conducted by Freney et al. (2011) at a mid-
altitude Puy-de-Dôme site and in a study conducted by Poulain et al. (2020) at a rural
background site in Melpitz. In this study, as the mass fraction of organics in the aerosols
increased, the density calculated using Eq. (2) converged to a value of 1.2 g cm$^{-3}$ (Fig. A14).
Table 5. Particle effective densities (g cm$^{-3}$) calculated during episodes of high mass
concentrations using AMS data.

| Episode AMS | S1 | S2 | S3 | S4 | S5 | S6 | S7 | S8 | S9 | S10 |
|---|---|---|---|---|---|---|---|---|---|---|
| Density[*] | 1.45 | 1.60 | 1.50 | 1.55 | 1.40 | 1.45 | 1.45 | 1.45 | 1.45 | 1.50 |
| Density[**] | 1.30 | 1.40 | 1.40 | 1.40 | 1.30 | 1.30 | 1.30 | 1.35 | 1.40 | 1.40 |
| # of spectra | 145 | 61 | 73 | 61 | 49 | 109 | 109 | 133 | 265 | 169 |


| Episode AMS | W1 | W2 | W3 | W4 | W5 | W6a | W6b | W7 | W8 | W9 | W10 | W11 | W12 | W13 |
|---|---|---|---|---|---|---|---|---|---|---|---|---|---|---|
| Density[*] | 1.40 | 1.40 | 1.70 | 1.60 | 1.70 | 1.6 | 1.55 | 1.55 | 1.60 | 1.45 | 1.75 | 1.50 | 1.60 | 1.55 |
| Density[**] | 1.40 | 1.50 | 1.50 | 1.50 | 1.50 | 1.50 | 1.40 | 1.30 | 1.30 | 1.30 | 1.50 | 1.40 | 1.40 | 1.40 |
| # of spectra | 175 | 229 | 337 | 85 | 25 | 805 | 307 | 19 | 25 | 19 | 97 | 115 | 31 | 139 |

[*] Density calculated using Eq. 1.
[**] Density calculated using Eq. 2 (Salcedo et al., 2006).

The differences between the densities obtained using the two approaches (spectra fitting – Eq.
1 versus chemical equation – Eq. 2.), ranging from 2 – 12% in summer and 7 – 19% in winter,
indicate the presence of different compounds of lower or higher densities that are not taken into
consideration by the effective density calculations, as well as the lower density used for
organics in Eq. (2), the physical characteristics of the particles, such as the particle size, porosity
and non-compactness, and calculation uncertainties that are primarily related to the single CE
correction used for the whole data set. The smaller differences between the two approaches
obtained in summer indicate aerosol particles composed mainly of NR-PM$_1$ species along with





eBC. In winter, the differences were larger, and both negative (compounds with lower densities and/or particle physical characteristics) and positive (compounds with higher densities) differences were obtained. However, the larger differences in winter could be strongly influenced by the considerable CE correction applied to the AMS data.

## 4. Summary and conclusions

This study is the first of its kind in the Czech Republic, assessing the seasonal variability of NR-PM$_1$ based on its chemically speciated mass size distribution, density, and origin at a rural background site. The impacts of atmospheric regional and long-range transport in Central Europe were examined based on intensive measurement campaigns conducted at National Atmospheric Observatory Košetice (NAOK) in summer 2019 and winter 2020.

The CE correction performed based on comparisons between sulphate concentrations measured by AMS and IC was applied to the NR-PM$_1$ data (0.4 in summer and 0.33 in winter), resulting in very good agreement between the AMS and MPSS volume and mass concentration in summer (slope=1.08, R$^2$=0.96 and slope=1.00, R2=0.97, respectively) and winter (slope=0.93, R$^2$=0.94 and slope=0.89, R$^2$=0.94, respectively). Near-real-time and systematic comparisons with reference methods represent the best way to obtain quality assurance of the AMS data and are needed to better characterize the robustness of the AMS data over long sampling time (Poulain et al., 2020).

The average NR-PM$_1$+eBC concentrations were 8.58±3.70 μg m$^{-3}$ in summer and 10.08±8.04 μg m$^{-3}$ in winter, with organics dominating during both seasons, followed by $SO_4^{2-}$ in summer and $NO_3^-$ in winter. The different seasonal compositions in PM$_1$ were caused by different sources and variable properties of individual compounds and were related to different meteorological conditions during these two seasons in the Czech Republic, as was previously mentioned by Kubelová et al., 2015.

The accumulation mode dominated the average mass size distributions during both seasons, with larger particles of all species in winter linked to seasonally differentiated regional and long-range origins as well as to the variability in the local sources primarily observed in winter. Although summer-aged continental air masses from the SE were rare (7%), they were connected to the highest concentrations of all NR-PM$_1$ species. In winter, the slow continental air masses from the SW (44 %) linked to inversion conditions over Central Europe were associated with the highest concentrations of organics, sulphate, nitrate, and ammonium.

The application of PMF on the PNSD enabled us to distinguish eight episodes of high particle contributions to N10-800 to calculate the particle effective density based on the particle number and mass size distributions. Additionally, a comparison of spectra fitting and chemical-based calculations for determining the particle effective density during episodes of high mass concentrations revealed differences in these two approaches due to the presence of compounds that were not taken into consideration by the density calculations, such as particle physical characteristics and calculation uncertainties.

*Data availability.*

All relevant data for this paper are archived at the ICPF of the CAS (Institute of Chemical Process Fundamentals of the Czech Academy of Sciences) and are available upon request from the corresponding author (Petra Pokorná).





*Author contribution.*
PP, JS and VŽ conceived the research. PP, RL, PV, SM, AHŠ and JO conducted the
atmospheric aerosol measurements during both intensive campaigns. PP, NZ, RL, PV, VR and
JS analysed and interpreted the data. PP prepared the manuscript with contributions from all
co-authors.
*Competing interests*
The authors declare that they have no conflict of interest.
*Acknowledgements*
We would like to thank Daniel Vondrák for the graphical editing. Thanks also goes to American
Journal Experts for the English proof reading of the manuscript. The authors gratefully
acknowledge the NOAA Air Resources Laboratory (ARL) for the provision of the HYSPLIT
transport and dispersion model and/or READY website (http://www.ready.noaa.gov) used in
this publication. We greatly thank the two anonymous reviewers for their effort to critically
review the manuscript and for providing constructive comments.
*Financial support*
This work was supported by the GACR under grant P209/19/06110Y and by the MEYS of the
Czech Republic under grant ACTRIS-CZ LM2018122 and ACTRIS-CZ RI
(CZ.02.1.01/0.0/0.0/16_013/0001315) as well as by COST Action CA16109 COLOSSAL
within STSM.

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



**APPENDIX**

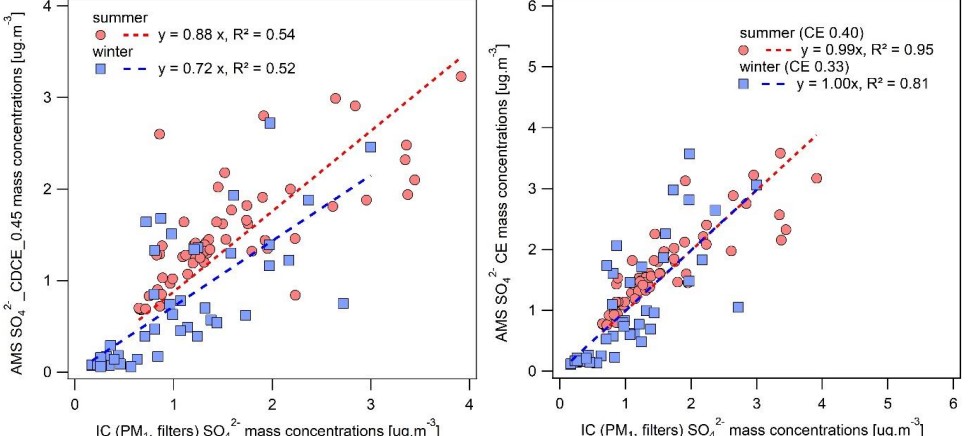


Figure A1. Comparison of sulphate concentration measured by AMS and retrieved from $PM_1$
filter analysis by IC with applied CDCE correction (left) and constant CE correction (right) for
both measurement seasons.


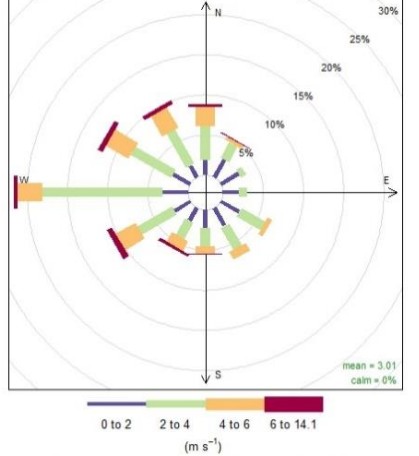
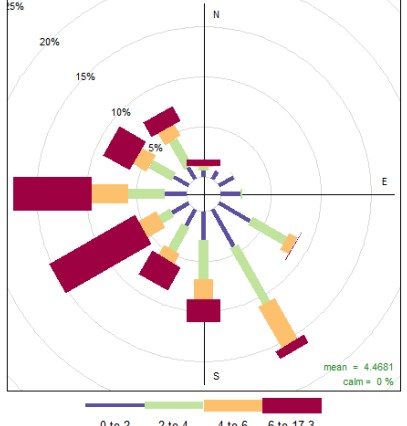


Figure A2. Wind rose summer (left) and winter (right).



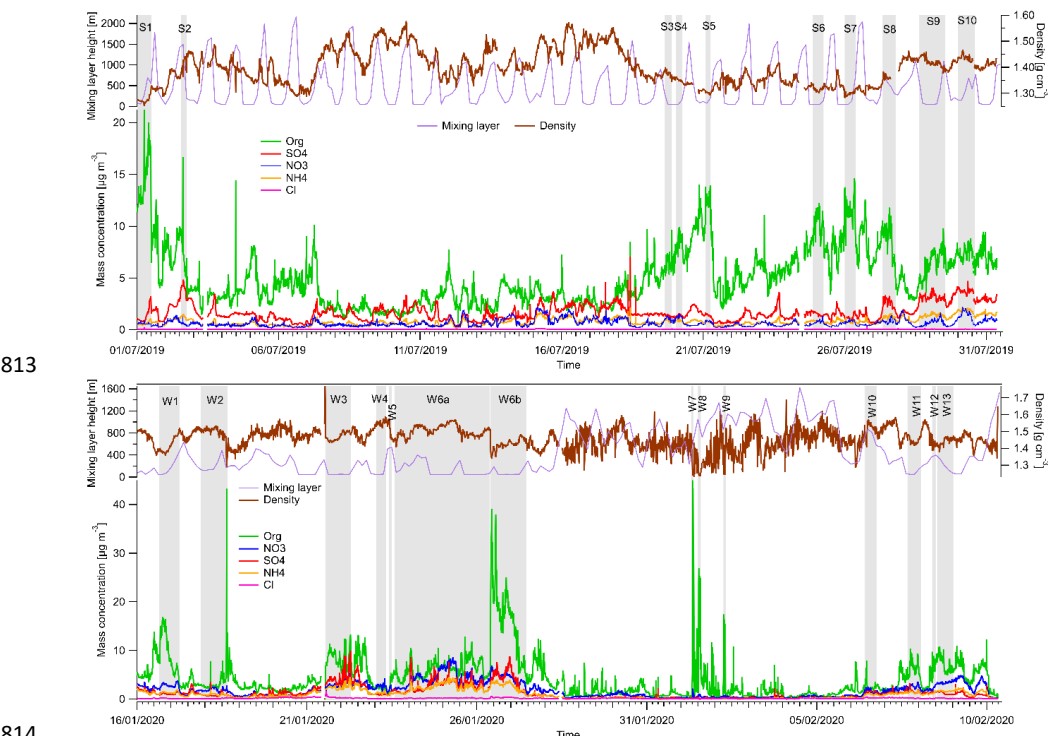



Figure A3. Mass concentration of Org, $NO_3^-$, $SO_4^{2-}$, and $NH_4^+$ measured by AMS with applied
constant collection efficiency (CE) correction for summer (top) and winter (bottom) campaign
with marked episodes of higher mass concentrations, mixing layer height and particle effective
density calculated using Eq. (2) in the main text from Salcedo et al., 2006.







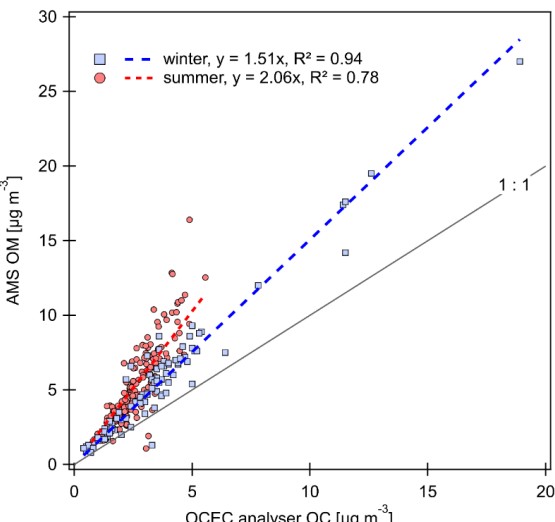


Figure A4. Comparison of organic mass concentration measured on-line by AMS (Org CE corrected) and by OCEC analyser in summer and winter.



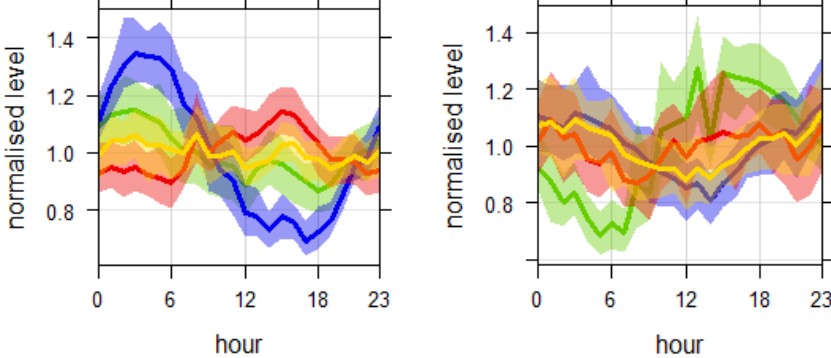


Figure A5. Diurnal trends of the NR-PM$_1$ species (common colour code) in summer (left) and winter (right).


Table A1. Overview table presenting mass (M) and median diameter (d) of NR-PM$_1$ species calculated by
fitting log-normal function to the AMS size distributions for the selected episodes in summer (S1 – S10) and
winter (W1 – 13) along with meteorology recorded during the episodes (relative humidity – RH, global
radiation – GR, temperature – T, wind speed – WS and wind direction – WD)

| Episode | Start | End | Duration [h] | M_Org [ug m$^{-3}$] | M_NO$_3^-$ [ug m$^{-3}$] | M_SO$_4^{2-}$ [ug m$^{-3}$] | M_NH$_4^-$ [ug m$^{-3}$] | d_Org [nm] | d_NO$_3^-$ [nm] | d_SO$_4^{2-}$ [nm] | d_NH$_4^-$ [nm] |
|---|---|---|---|---|---|---|---|---|---|---|---|
| S1 | 7.1.19 0:00 | 7.1.19 12:00 | 12 | 14.58 | 0.82 | 1.24 | 0.91 | 314 | 285 | 414 | 498 |
| S2 | 7.2.19 13:00 | 7.2.19 18:00 | 5 | 6.33 | 0.49 | 4.70 | 1.52 | 307 | 304 | 325 | 335 |
| S3 | 7.19.19 15:00 | 7.19.19 21:00 | 6 | 6.71 | 2.00 | 1.84 | 1.15 | 373 | 421 | 470 | 453 |
| S4 | 7.20.19 1:00 | 7.20.19 6:00 | 5 | 8.41 | 2.03 | 1.58 | 1.21 | 365 | 388 | 467 | 466 |
| S5 | 7.21.19 2:00 | 7.21.19 6:00 | 4 | 10.83 | 1.01 | 1.53 | 0.95 | 358 | 333 | 473 | 504 |
| S6 | 7.24.19 21:00 | 7.25.19 6:00 | 9 | 8.94 | 0.97 | 1.59 | 1.07 | 284 | 271 | 366 | 412 |
| S7 | 7.26.19 0:00 | 7.26.19 9:00 | 9 | 9.25 | 0.98 | 1.43 | 0.99 | 279 | 253 | 382 | 454 |
| S8 | 7.27.19 8:00 | 7.27.19 18:59 | 10 | 9.63 | 1.36 | 3.54 | 1.56 | 399 | 412 | 439 | 436 |
| S9 | 7.28.19 15:00 | 7.29.19 13:00 | 22 | 6.78 | 1.16 | 4.49 | 1.76 | 409 | 414 | 430 | 439 |
| S10 | 7.30.19 0:00 | 7.30.19 14:00 | 14 | 9.57 | 3.37 | 6.14 | 2.98 | 466 | 491 | 494 | 478 |
| W1 | 1.16.20 15:30 | 1.17.20 6:00 | 14.5 | 8.60 | 5.63 | 1.39 | 3.47 | 357 | 378 | 447 | 392 |
| W2 | 1.17.20 21:00 | 1.18.20 16:00 | 19 | 4.04 | 5.84 | 1.45 | 3.83 | 356 | 428 | 456 | 429 |
| W3 | 1.21.20 13:00 | 1.22.20 17:00 | 28 | 9.33 | 7.50 | 7.13 | 7.90 | 563 | 609 | 636 | 607 |
| W4 | 1.23.20 1:00 | 1.23.20 8:00 | 7 | 1.90 | 7.04 | 1.89 | 4.48 | 388 | 386 | 487 | 410 |
| W5 | 1.23.20 10:00 | 1.23.20 12:00 | 2 | 4.26 | 7.27 | 3.20 | 5.46 | 357 | 386 | 433 | 391 |
| W6 | 1.23.20 14:00 | 1.27.20 11:00 | 93 | 7.82 | 9.40 | 4.18 | 6.76 | 460 | 586 | 630 | 588 |
| W6a | 1.23.20 14:00 | 1.26.20 9:00 | 67 | 6.18 | 10.66 | 4.15 | 7.55 | 523 | 584 | 629 | 584 |
| W6b | 1.26.20 9:30 | 1.27.20 11:00 | 25.5 | 13.23 | 6.37 | 4.34 | 4.89 | 398 | 571 | 625 | 593 |
| W7 | 2.1.20 7:30 | 2.1.20 9:00 | 1.5 | 15.63 | 0.93 | 0.74 | 0.96 | 336 | 276 | 241 | 390 |
| W8 | 2.1.20 12:00 | 2.1.20 14:00 | 2 | 10.32 | 0.72 | 0.62 | 0.90 | 295 | 240 | 242 | 365 |
| W9 | 2.2.20 6:00 | 2.2.20 7:30 | 1.5 | 10.12 | 0.17 | 0.41 | 0.76 | 296 | 787 | 287 | 392 |
| W10 | 2.6.20 10:00 | 2.6.20 18:00 | 8 | 2.15 | 2.66 | 4.19 | 3.35 | 385 | 479 | 473 | 462 |
| W11 | 2.7.20 16:00 | 2.8.20 1:30 | 9.5 | 5.76 | 5.09 | 2.50 | 3.30 | 366 | 419 | 488 | 446 |
| W12 | 2.8.20 9:30 | 2.8.20 12:00 | 2.5 | 6.52 | 5.23 | 2.27 | 3.06 | 387 | 461 | 523 | 478 |
| W13 | 2.8.20 13:00 | 2.9.20 0:30 | 11.5 | 7.72 | 8.12 | 1.93 | 4.35 | 379 | 436 | 498 | 451 |

| Episode | Start | End | Duration [h] | RH [%] | GR [W m$^{-2}$] | T [°C] | WS [m s-1] | WD |
|---|---|---|---|---|---|---|---|---|
| S1 | 7.1.19 0:00 | 7.1.19 12:00 | 12 | 49 | 318 | 25.8 | 3.7 | W-SW |
| S2 | 7.2.19 13:00 | 7.2.19 18:00 | 5 | 44 | 566 | 22.8 | 3.7 | N-NNW |
| S3 | 7.19.19 15:00 | 7.19.19 21:00 | 6 | 91 | 92 | 17.3 | 1.5 | S-SE-W |
| S4 | 7.20.19 1:00 | 7.20.19 6:00 | 5 | 97 | 28 | 14.9 | 1.3 | SE |
| S5 | 7.21.19 2:00 | 7.21.19 6:00 | 4 | 68 | 31 | 19.7 | 2.5 | SW-NW |





| | | | | | | | | |
|---|---|---|---|---|---|---|---|---|
| S6 | 7.24.19 21:00 | 7.25.19 6:00 | 9 | 68 | 13 | 18.2 | 1.2 | SW-SE |
| S7 | 7.26.19 0:00 | 7.26.19 9:00 | 9 | 59 | 148 | 19.1 | 2.3 | W |
| S8 | 7.27.19 8:00 | 7.27.19 18:59 | 10 | 75 | 297 | 21.3 | 3.4 | SE |
| S9 | 7.28.19 15:00 | 7.29.19 13:00 | 22 | 81 | 156 | 20.5 | 2.4 | W-NW-SE |
| S10 | 7.30.19 0:00 | 7.30.19 14:00 | 14 | 81 | 196 | 20.9 | 3.7 | W |
| W1 | 1.16.20 15:30 | 1.17.20 6:00 | 14.5 | 92 | 3 | 1.1 | 2.1 | SE |
| W2 | 1.17.20 21:00 | 1.18.20 16:00 | 19 | 96 | 13 | 0.4 | 2.0 | SE-NW |
| W3 | 1.21.20 13:00 | 1.22.20 17:00 | 28 | 93 | 77 | -3.8 | 2.5 | W-NW-SW |
| W4 | 1.23.20 1:00 | 1.23.20 8:00 | 7 | 88 | 0 | 0.1 | 1.7 | W-NW |
| W5 | 1.23.20 10:00 | 1.23.20 12:00 | 2 | 73 | 120 | 0.6 | 1.9 | SE |
| W6 | 1.23.20 14:00 | 1.27.20 11:00 | 93 | 93 | 34 | -1.1 | 1.7 | SE-S-SW |
| W6a | 1.23.20 14:00 | 1.26.20 9:00 | 67 | 94 | 20 | -2.4 | 2.0 | SE-S |
| W6b | 1.26.20 9:30 | 1.27.20 11:00 | 25.5 | 98 | 43 | -1.0 | 1.1 | SE |
| W7 | 2.1.20 7:30 | 2.1.20 9:00 | 1.5 | 77 | 22 | 9.2 | 3.9 | SW |
| W8 | 2.1.20 12:00 | 2.1.20 14:00 | 2 | 69 | 201 | 11.9 | 7.5 | SW |
| W9 | 2.2.20 6:00 | 2.2.20 7:30 | 1.5 | 75 | 0 | 4.1 | 8.1 | W |
| W10 | 2.6.20 10:00 | 2.6.20 18:00 | 8 | 76 | 112 | 0.4 | 6.0 | W-NW |
| W11 | 2.7.20 16:00 | 2.8.20 1:30 | 9.5 | 92 | 4 | 0.9 | 1.5 | SE |
| W12 | 2.8.20 9:30 | 2.8.20 12:00 | 2.5 | 85 | 237 | 0.8 | 3.9 | SE |
| W13 | 2.8.20 13:00 | 2.9.20 0:30 | 11.5 | 84 | 86 | 0.6 | 2.7 | SE |



Figure A6. Backward air mass trajectories calculated by HYSPLIT for corresponding summer
episodes (S1 – S10) of high concentration of species size distributions.



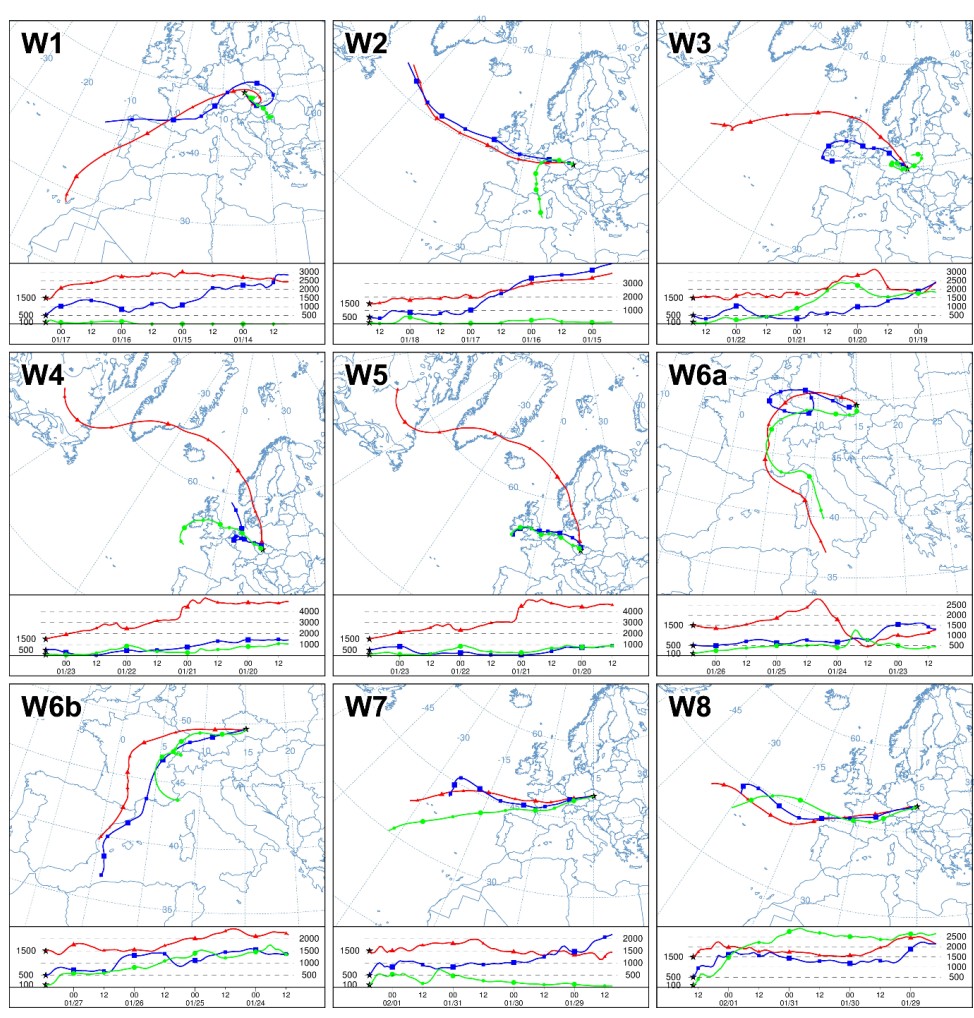


Figure continues.





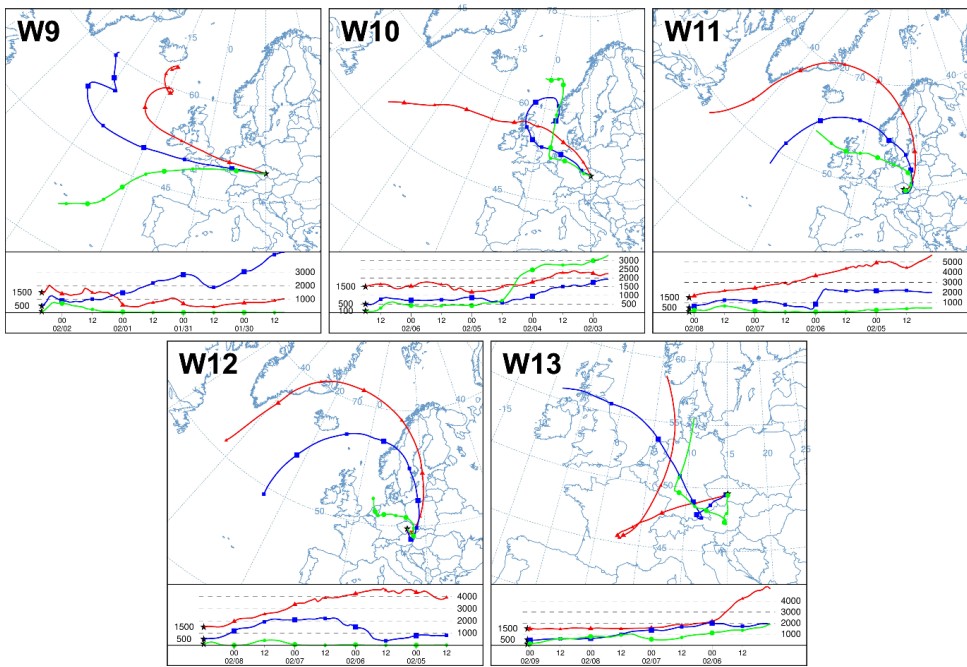

Figure A7. Backward air mass trajectories calculated by HYSPLIT for corresponding winter episodes (W1 – W13) of high concentration of species size distributions.

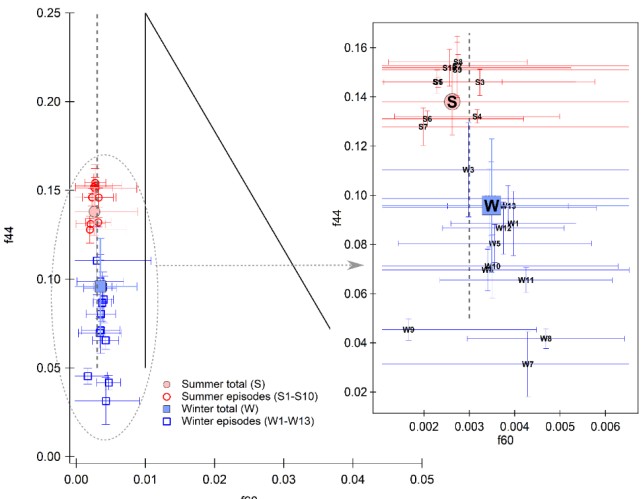

847

Figure A8. Comparison of organic fragments $f_{44}$ and $f_{60}$ for the whole campaigns (full markers)
and for the specific episodes (empty markers). Bars represent standard deviation and the
triangular space area of biomass burning (BB) influence and dashed line a limit for a negligible
fresh BB influence (Cubison et al., 2011).


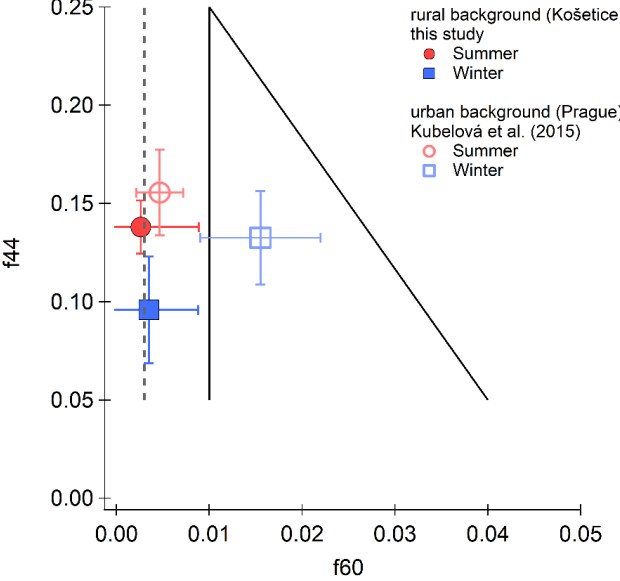


Figure A9. Comparison of organic fragments $f_{44}$ and $f_{60}$ determined at rural background site
(NAOK) and urban background site (Prague, study by Kubelová et al., 2015) during summer
and winter seasons.







Figure A10. PNSD factor profiles for summer (top) and winter (bottom) campaign. The bars
represent the number size distribution (y-axis on the left), and the lines represent the
contribution as a percentage (y-axis on the right).






Table A2. Summary of PMF diagnostics for PNSD.

| Diagnostic | Summer | Winter |
|---|---|---|
| N. of observations | 8684 | 7414 |
| Missing values | 6.8% | 0% |
| Number of factors | 6 | 5 |
| $Q_{expected}$ | 161224 | 103701 |
| $Q_{true}$ | 129774 | 102925 |
| $Q_{robust}$ | 130657 | 103495 |
| Species with $Q/Q_{expected}>2$ | 0 | 263 |
| Extra modelling uncertainty | 4.8% | 4.0% |
| DISP swaps | 0 | 0 |
| BS mapping | 100% | 100% |







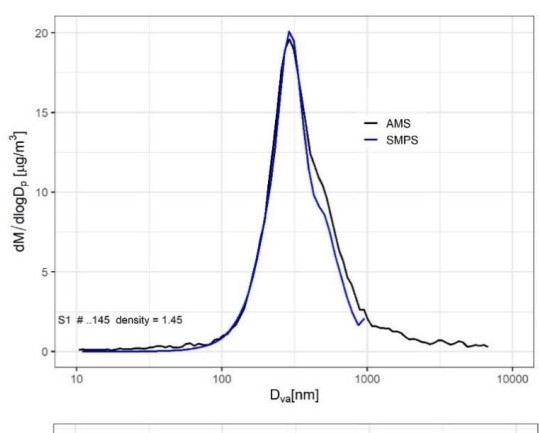


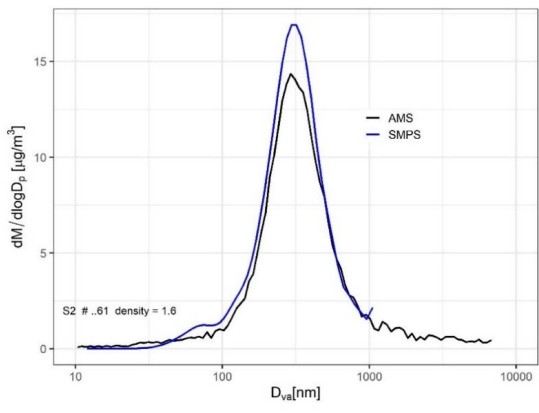


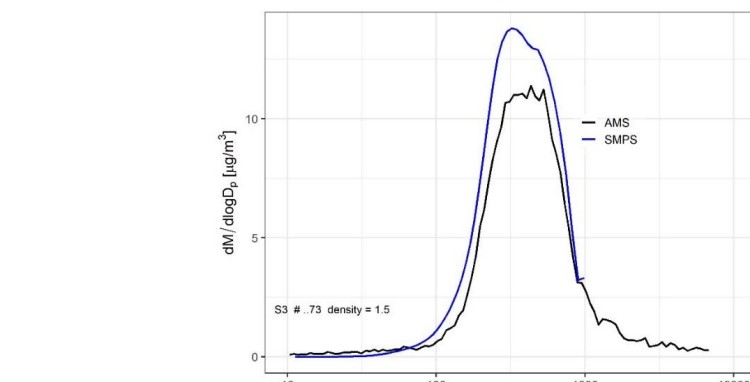





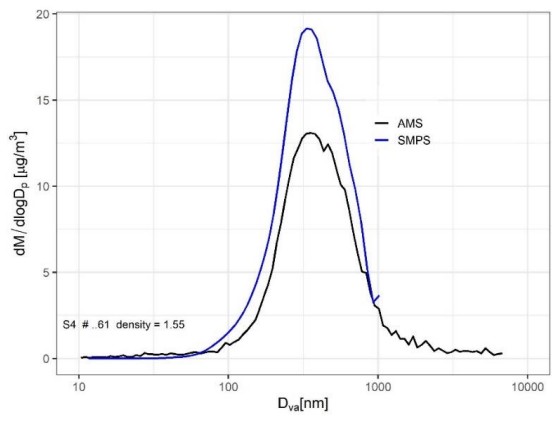


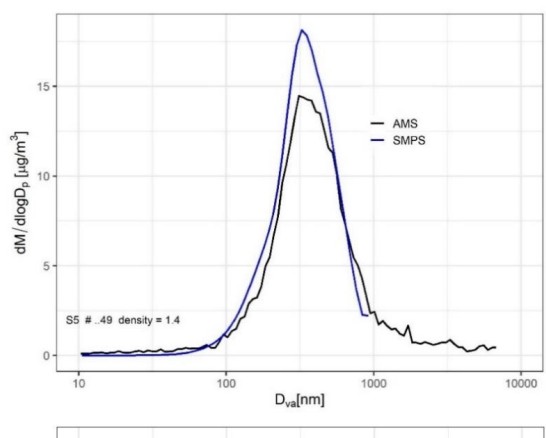


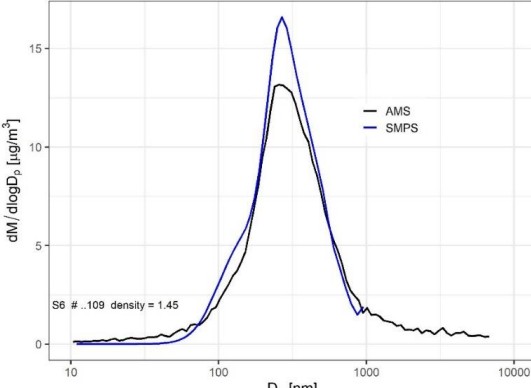






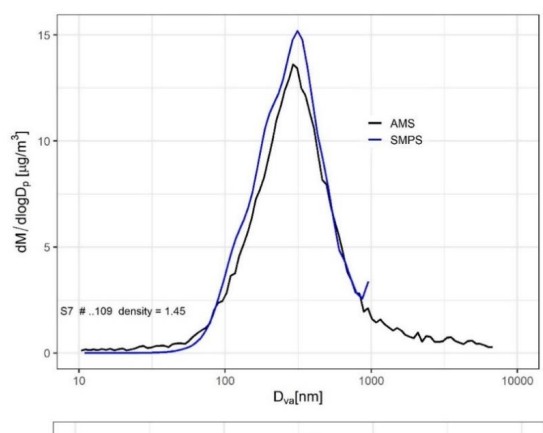


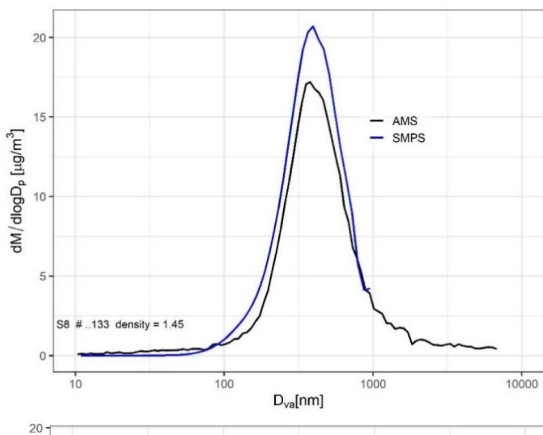


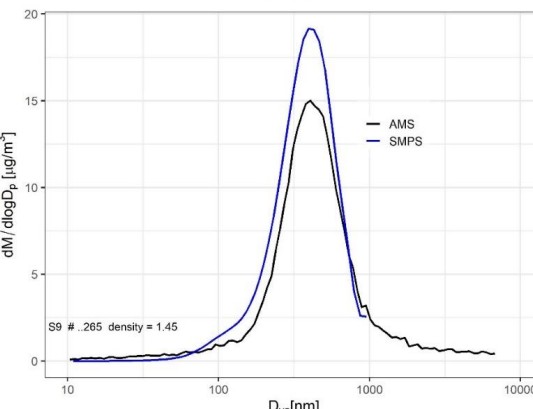




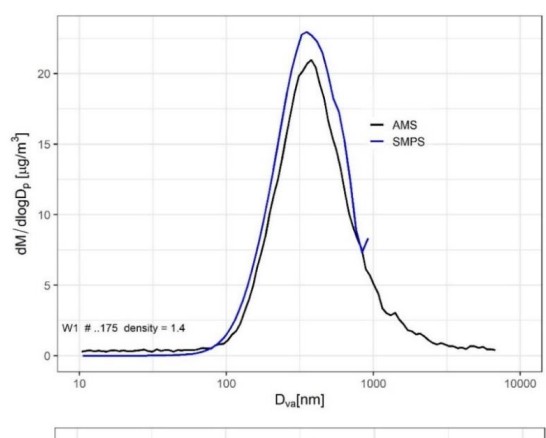



Figure A11. Fit of AMS and MPSS mass size distribution spectra of summer episodes (S1 –
S10) for density calculation.

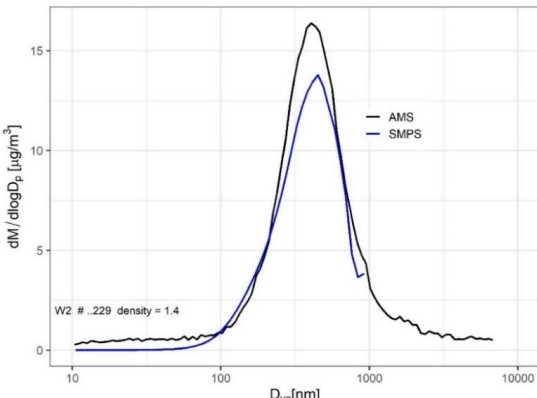





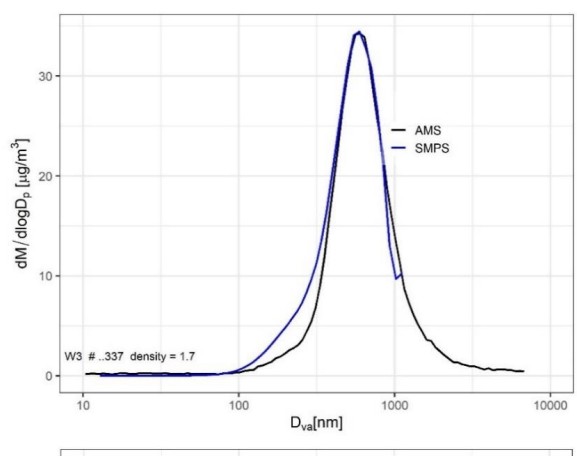


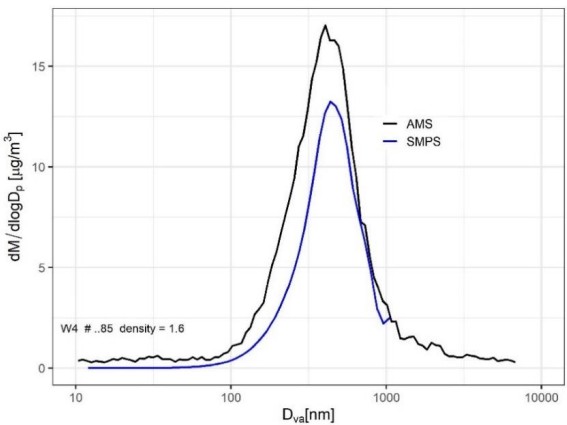


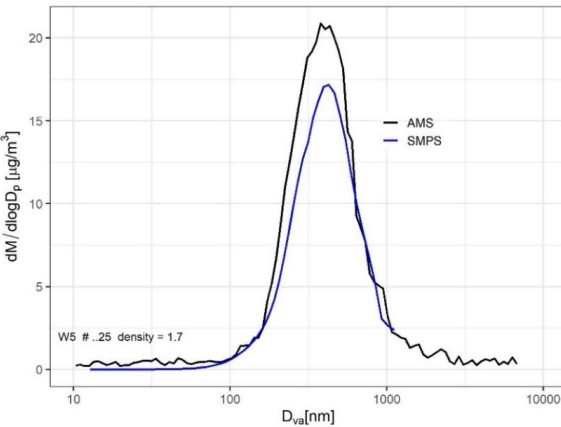






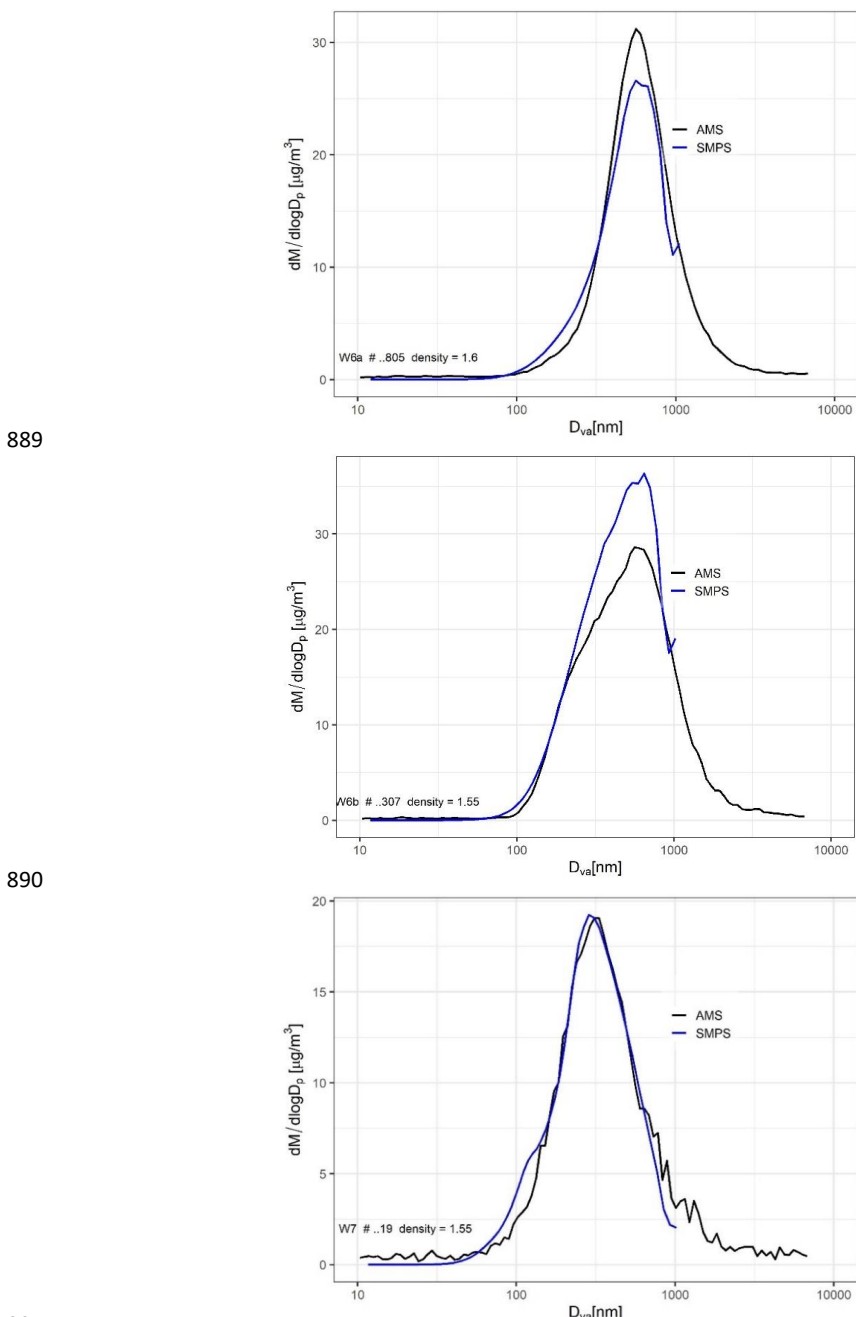






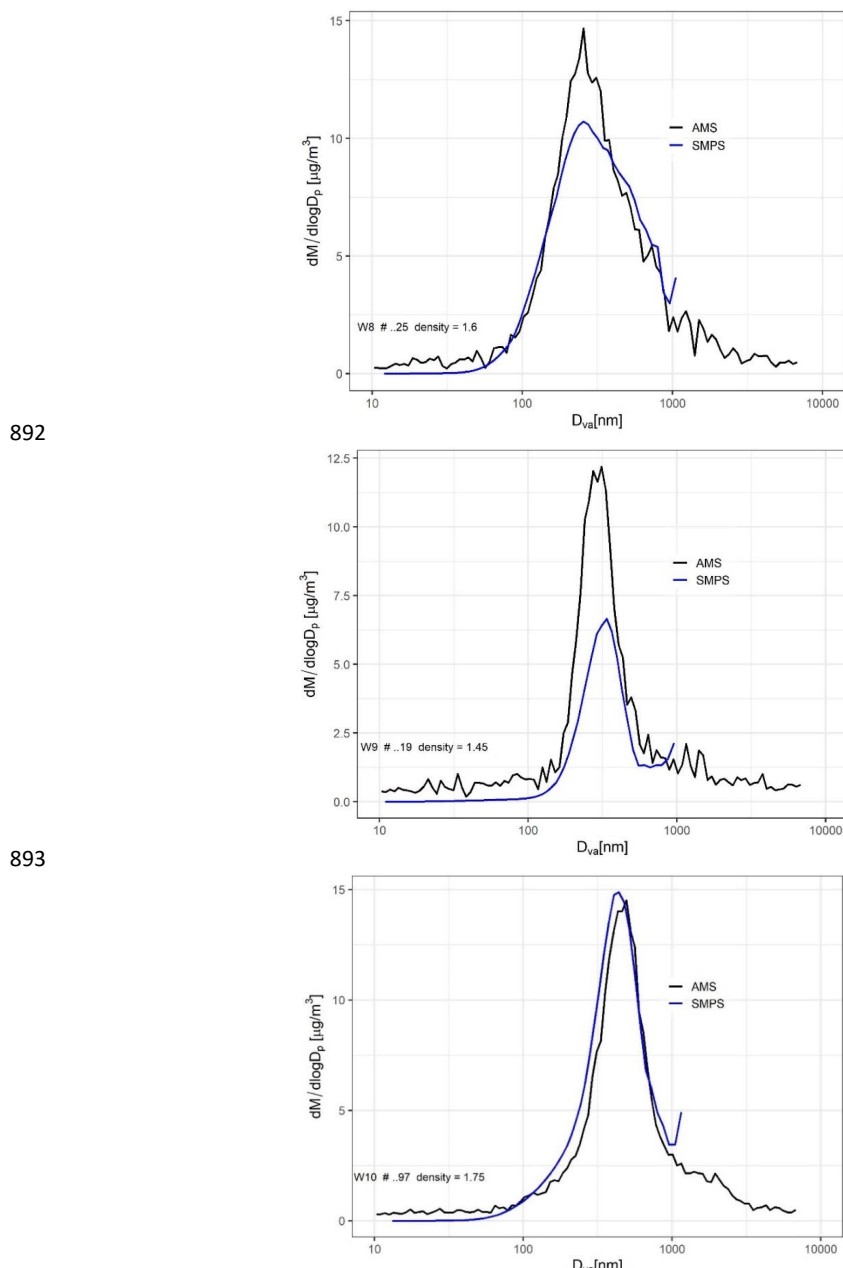











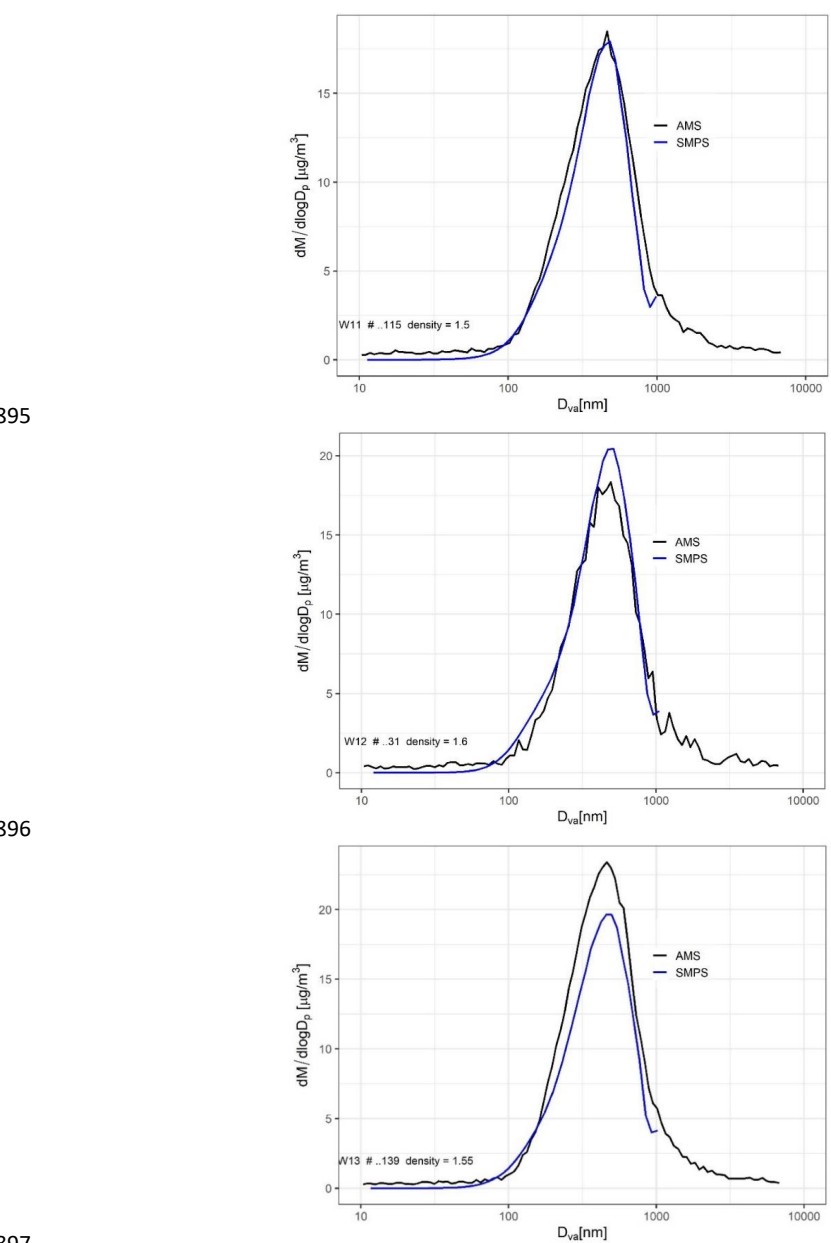


Figure A12. Fit of AMS and MPSS mass size distribution spectra of winter episodes (W1 –
W13) for density calculation.





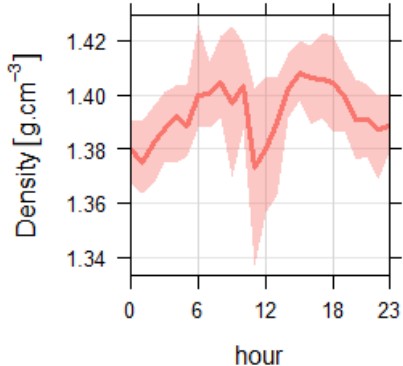 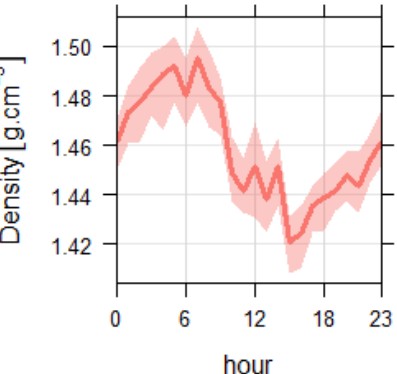

Figure A13. Diurnal trends of average effective particle density calculated based on Eq. (2) in
the main text from Salcedo et al., 2006 in summer (left) and winter (right).














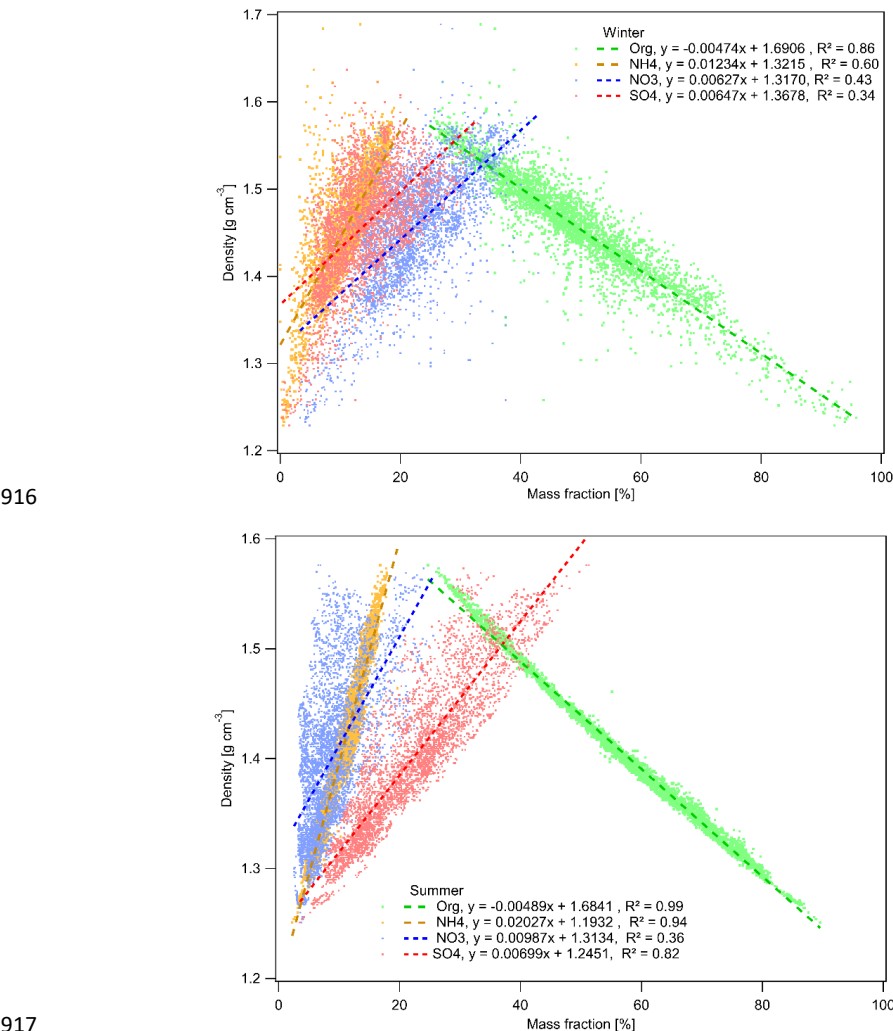


Figure A14. Relationship between density, calculated according to Eq., and mass fractions of
the main NR-PM$_1$ species.

