# Peer review of "Chemically speciated mass size distribution, particle density, shape and origin of non-refractory PM1 measured at a rural background site in Central Europe"

_Atmospheric Chemistry and Physics, 2021_

## Author Comment (AC1)

We would like to thank the reviewer for the positive and constructive comments that helped to improve the present study. Responses to each comment are provided below - see blue text.

**REVIEW #1**

Review to „Chemically speciated mass size distribution, particle effective density and origin of non-refractory PM1 measured at a rural background site in Central " by P. Pokorna et al.

This manuscript reports on two one-month field measurements conducted in summer and winter, respectively. This study is the first of its kind in the Czech republic. The results are used to infer general particle properties, the origin of the aerosol particles, and their effective density of the particles.

The fact that this study is the first in the Czech republic makes the data set interesting, but on the other hand, Czech Republic is in the middle of Europe, so one would not expect to see large differences in chemical composition and particle origin compared to neighboring countries where many studies like this (using ACSM and AMS instrument) have been already conducted.

Furthermore, the results are not presented in a clear and concise way. There is no clear concept in this manuscript, no "story line". The analyses are done one by one, without connecting them, and many times the order of the analyses is confusing to the reader. The overall findings are not clearly summarized and highlighted. Some weaknesses are found in the discussion of the effective density.

I therefore recommend major revisions according to my comments below

**Major overall comments**

1) Effective density

The authors use two different equations to calculate the effective density. The first is taken from DeCarlo et al. (2004). Here it should be mentioned that DeCarlo et al. give three different possible definitions of the effective density. This should be commented.

The second definition used here relies on the measured composition of the particles. Thus, this represents the "real" density of the particle material, in DeCarlo this is denoted with "rho_m". This is not an effective density. In fact, using DeCarlo's equation 45, one can infer the Shape factor from measuring rho_eff and rho_m. Thus, I think here is some general lack of understanding what the measured quantities mean.

Thank you very much for this valuable comment. This mistake was corrected. In Section 2.2.2 the densities, effective density ($\rho$eff) and material density ($\rho_m$) were defined as well as the Jayne Shape factor (S) and the dynamic shape factor ($\chi$). In Section 3.5 and 3.6 the results for the densities and shape factors are discussed.

**2) Structure of the manuscript**

As I mentioned above, there is no clear story line in the manuscript. It more reads like a measurement report.

For example I suggest to discuss the results from the clustered trajectory analysis before the single events. The presentation should progress from a general overview (Fig 2) to the general results of air mass origin (Fig 4) to the specific events (Fig 3 + Fig 5).

The authors have tried to make the manuscript more readable and with clear story line.

Furthermore, the definition of the events needs to be made clear. I also suggest moving a few figures from the appendix to the main text (see details below). Tables, in contrast, may preferably go into the appendix.

The criteria for episodes of high mass concentrations were set. The suggested exchanges of the figures were done.

3) PMF of particle number size distributions

I am not sure about the PMF results. Factors obtained by PMF have to be checked whether they are physically meaningful. I don't see how this has been done here. The presented factors do not look like ambient aerosol number size distributions. This needs to be justified and better explained.

PMF of PNSD is described in Section A1 (in Appendix) along with two new figures A12 and A13.

**Specific comments**

Lines 138-139:

Is this only the sulfate CE? What CE values were used for the others?

Yes, the final correction of AMS data is based on sulphate CE. This is because the AMS data was underestimated compared to the filters when using the CDCE correction procedure (see Fig. A1 left). The underestimation of the CDCE corrected data was due to the fact that the CDCE algorithm assumes a threshold CE of 0.45 in this correction. However, as the comparison with the filters shows the CE on our instrument was lower - 0.4 for summer and 0.33 for winter as imply from the sulfate comparison (see Fig. A1 right). We then used these CE values for other species, although this may reflect some imprecision. However, as the OM vs. OC comparison (Fig. A4) shows, the AMS OM data corrected in this way seem realistic.

The same method of correction was successfully used in a previous measurement with this C-ToF-AMS instrument, where the CE values were also low – see Kubelová et al. (2015).

Lines 150 - 153:

I don't understand in the first approach how you calculated Dm. Why use 1.5 g/cm3? Where does this come from? Why not calculating dV from dN (using D_m), and then comparing the main mode of MPSS and AMS and adjusting the density (thereby changing D_va to D_m with Equ.1) until the position of the main modes match?

Thank you for noticing this – it is an artifact of the data processing. The procedure for density calculation is exactly the one you described; however the AMS data were not in the original D_va but calculated to D_m with a density of 1.5. To remove this step (not used in the final analyses), it was recalculated back to D_va. To remove the confusion resulting from notes among co-authors rather than a proper description of the data processing, the paragraph was reformulated to:

DeCarlo et al. (2004) give three different possible definitions of the effective density estimation: i) from mobility and mass measurements, ii) as fitted parameter, and iii) from mobility and aerodynamic measurements. Here we proceed from the latter definition with the AMS data representing the mass

size distributions based on the vacuum aerodynamic diameter ($D_{va}$) in the size range approximately from 50 to 800 nm (in Squirrel software extrapolated in range $10-7000$ nm), and MPSS data based on mobility diameter ($D_m$) representing the dN/dlog $D_p$ in the size range from 11.3 to 987 nm. In the MPSS data, the $D_m$ were recalculated to vacuum aerodynamic diameters with the assumption of spherical particles as in DeCarlo et al. (2004):

$$D_{va} = \frac{D_m}{\rho_0}\rho,\qquad\qquad\qquad(1)$$

where $D_m$ is the mobility diameter, $D_{va}$ is the vacuum aerodynamic diameter, $\rho_0$ is the water density, and $\rho$ is the total density of particles

The position of the main mode of mass distribution was compared between the AMS and MPSS data to estimate the aerosol effective density.

Line 167:

Is the organic density from Turpin and Lim appropriate for your conditions?

We used the density value for organic matter from the study of Turpin and Lim (2001) (1.20 g cm$^{-3}$) deliberately because it is widely used at various stations, including background stations. However, as our results suggest, this widely accepted value does not match the conditions at our background site. This is discussed in Section 3.5.2 and it is also one of the conclusions of this study, with the implication that used value of density of the organic matter should generally be higher at background sites.

Lines 184 ff (Section 2.2.4.1):

I don't understand what was chosen here. What was the intention? To find out episodes with highest number concentration? Why do you need PMF for that? Why not taking the time series of the total number concentration and select based on a threshold or based on probability density function?

PMF was applied to PNSD to estimate the PNSD factors contributions to the receptor. This enables us to obtain episodes with high number concentrations (criteria - factor contributions to N10-800 >80%) of one factor and therefore of same origin reflected as well as in the particle density. The eight winter episodes were linked to two factors: F3, mode diameter ~429 nm, $\rho\_eff$ 1.85 g cm$^{-3}$, $\rho\_m$ 1.50 g cm$^{-3}$ – N_W1 and F1, mode diameter ~32 nm, $\rho\_eff$ $1.40-1.60$ g cm$^{-3}$, $\rho\_m$ $1.30-1.55$ g cm$^{-3}$ – N_W 2 – 8.

In this paper, the PMF represents an auxiliary analysis to obtain high number concentrations episodes for density calculation.

Lines 231-232:

Why are you using a constant density and not the density derived from AMS?

Since the calculated density is AMS mass depended we hesitated use it for the PNSD conversion into mass concentration by AMS-MPSS mass closure.

Lines 233-239:

Shouldn't the transmission effects be taken into account by the CE correction? As far as I understood, the CE correction was based on comparison of AMS data to filter data. So the same method as applied

here. So, why is there a difference? This also refers to my question on the CE values for the other species above.

Yes, the reviewer is right. The CE correction takes in account the transmission effect. The CE correction is discussed in detail by other reviewer's comments. Since just the volume closure results were presented in the paper manuscript, the whole paragraph was deleted.

Lines 240-241:

Yes, the constant density is certainly a limitation. So why not use the density inferred from the chemical composition? Or otherwise: Why presenting two closures? For volume closure, you convert AMS mass to volume using the measured density, for mass closure, you convert the MPSS volume to mass using a constant density of 1.5. What additional insights do you expect from the mass closure if it is clear that the assumed density is a simplification?

The reviewer is right. Although it is a common approach of many studies to convert the PNSD into mass concentration using constant density we have decided to present just the AMS-MPSS volume closure.

Lines 254-256:

Are the CE estimation and the mass closure also affected from this NH4NO3 loss? I would think so.

Yes, you are correct that $NH_4NO_3$ losses at the filters can affect the total mass of PM1 and hence the CE is corrected through the total mass. This is a reason why we corrected for CE by comparing it with sulfate concentrations (see subsection 2.2.1) and not by a whole filter mass.

The CE correction, based on sulfate concentrations comparison, in this case provided more relevant results than CDCE correction (see Fig.A1). This is probably due to the fact that the CE on our AMS was lower than the threshold value of 0.45 used in CDCE corrections.

The CE correction factor based on sulfate comparisons also provided relevant results when applied to the total organic matter (see Fig.A4). We conclude that the CE correction factor used in this way will also be suitable for other chemical constituents measured by AMS ($NH_4^+$, $NO_3^-$).

Table 2:

Why do you calculate the NR-PM1 share? Why not taking AMS +eBC as the "total PM1" reference? I think we would learn more from a "share on PM1".

The share on PM1 was calculated and is presented in the Table 2.

Line 306 (caption Fig 2):

Again, why not taking the measured density?

For the easier comparison of the particles sizes (number and volume size distributions by MPSS and mass size distributions by AMS) the mobility diameters of MPSS data were recalculated to aerodynamic diameters. However, the density was not taken into account.

Line 308:

What were the exact selection criteria? The explanation given in 2.2.4 is not clear. Additionally I think it would be helpful to indicate the selected events in Fig 2.

The criteria for the episode selection are described in the Section 2.2.4: … episodes of high mass concentrations were chosen based on a set of criteria: high mass size distribution of at least one main NR-PM1 specie corresponding to the season ($NO_3^-$ ≥ 0.5/0.2 µg m$^{-3}$, $SO_4^{2-}$ ≥ 1/0.5 µg m$^{-3}$, $Org$ ≥ 6 /2 µg m$^{-3}$); monomodal mass size distribution of all main NR-PM1 species; duration of the episodes min 1.5 hour. Also, the episodes were marked as well in Figure 2.

Line 311 ff, Figs A6, A7, and Section 2.2.3:

It is not clear how the trajectories were calculated. In Section 2.2.3 it is said „500 m". In Fig A6 and A7 it appears that 100, 500, and 1500 m were used.

Trajectory analysis requires more information on variability. Either initialize the calculate at least every hour during the selected events (some events last more than 12 hours, a time during which the air mass origin may change), or use the "ensemble" mode where HYSPLIT varies the initial conditions. This allows for a better estimation on the spread and the variability of the air mass movements

The Section 2.2.3 was reworded and figures (Fig A5 and A6) redone. The trajectories for episodes of high mass concentrations were recalculated with a 500 m AGL position and the trajectory ensemble option with calculation initialized every hour and duration 72-h was utilized.

Line 330:

What is f44f60? Furthermore, f44 and f60 have not been introduced up to here.

The organic fragments were introduced in text first and used later as f44 and f60.

Lines 335 ff:

Please give a short explanation what f43 and f44 are. Not every reader is an AMS user. The paragraph needs references (e.g. to Ng et al., 2010, for f43 and f44), but also later when MOOA and LV-OOA are mentioned (e.g. to Jimenez 2009, Crippa 2013 etc).

Added to manuscript: "Organic fragments f44 and f43 (ratios of organics in m/z 44 and m/z 43 to total organics) can serve as a proxy of aerosol oxidation and its aging, respectively (Ng et al., 2010). In simplified form, more oxidized aerosols have higher f44 and lower f43 while less oxidized and more volatile aerosols have the opposite f44 vs f43 relationship. These oxidation properties of organic aerosols are well defined by the triangular region defined by Ng et al. (2010). This triangular area is shown in Fig. 5 together with the evolution of f44 and f43 fragments during both campaigns."

Lines 347 ff:

I am surprised by the low BB content in the rural winter data. Can you comment on lifetime and aging of f60? See Cubison et al. (2011), Hennigan et al (2011) and Milic et al. (2017)

This is one of the interesting results of this study. In fact, the rural station is not low in BB in winter as indicated by the levoglucosan concentration data, but is low in f60. Comparison of rural and urban

sites data (Fig. A9) shows that the average f60 values are already less useful as an indicator of BB at the background station where aerosols are more aged.

Fig 4:

The bars for chloride in winter look strange. Were there many data gaps or data below the detection limit? Please explain.

There was much BDL data in winter, therefore, the boxplot for chloride was excluded.

Fig. 4:

Please add total PM1, eBC and total particle number to Fig 4.

The variables were added to the Fig. 4.

Lines 395-396:

The episodes need to be defined earlier (see comment above asking for the selection criteria), and this would include also the split of W6 episode.

The episodes were defined in Section 2.2.4 along with the criteria.

Table 3:

Caption should read "Mode diameter of mass distributions"

We changed text in caption from "Average size distribution…"to "Mode diameter of mass distributions…"

Lines 440-442 and Fig 5:

This finding is interesting and deserves more discussion. The correlation between organic particle size and f44 is a new result. Is the correlation for summer significant? Do you expect a linear relationship? Have there been other studies doing this?

Thank you. We do not know of any other studies that report a dependence of f44 on organic particle size. Both correlations are significant, p-value for summer values is 1.6e-8. The linear fit was performed as the simplest analysis to show the strength of the correlation. However, it is likely that the dependence will be linear within a certain size range in reality. To prove this, however, we would need more measurements over a larger size range than our AMS provides.

For Figure 6, we have expanded the text in the discussion as follows (see red text):

Additionally, as expected,The aging of aerosol particles is often connected with particle growth similarly as with oxidation of organic mass. Comparison of fragment f44 content withthe Org particle size showed growth, and the increasing mode diameter fully confirmed the ideas. was more significant in the winter season, with the ageing of aerosols resulting in oxygenated organic aerosols (Fig. 56). In both seasons, the correlation of the linear fit between Org size and f44 iswas significant (p-value < 0.001). However, the data presented here does not allow us to extend this size range due to both

instrumental (C-ToF-AMS particle size range is ca from 50 – 800 nm) and data characterization reasons, as we did not observe a major mode of organics at sizes below 200 nm.

In general, however, Fig. 6 suggests that the larger the particles with the organic contribution, the more oxidised they are due to its longer residence time in the atmosphere. The milder slope of the line for the summer dataset (Fig. 6) indicates that oxidation is still occurring on the particles, but appears to be approaching somean oxidation limit with growing particle size.  In the case of winter, the steeper slope of the line and lower f44 values for smaller particles suggest that the level of oxidation ischange of oxidation state with particle size is relatively more intense than in summer (Fig. 6). However, even so, under the given winter conditions (e.g., lower photochemical oxidation in winter than in summer), the degree of oxidation of organic aerosols does not reach the same level as in summer.

Lines 449 ff (Section 3.5):

Are the episodes here the same as S1-10 and W1-13?

There are two sets of episodes, episode of high particle numbers (N_W1 – N_W8) discussed in Section 3.5.1 and mass concentrations (S1 – S10 and W1 –W13) discussed in Section 3.4 and 3.5.2.

Lines 453-454:

PMF: How did you decide what is the most physically meaningful result? The whole PMF procedure need more explanation. Is it common to apply PMF to PNSD? Are there references?

The PMF procedure is explained in Section A1 (in Appendix). Most physically meaningful result means factor profiles of lognormal distribution (Fig. A12) and no artificial factor/s (e.g. split of two factors) assessed based on factor origin (e.g. polar plots).

Yes, it is common to apply PMF to PNSDs.  Following text with references were added to Section 2.2.4.1 PMF on PNSD: Application of PMF on PNSD is commonly adopted in source apportionment studies since by investigating particles in various size ranges, it is possible to more clearly identify and apportion contributions from those sources that contributed more to the particle number than to the particle mass (e.g. Beddows et al., 2015; Masiol et al., 2016; Sowlat et al., 2016; Leoni et al., 2018; Pokorná et al., 2020; Zíková et al., 2020).

What are "two-sided size bins"? Each bin has two sides. Do you mean "the first two bins and the last two bins"? And did your downweigh them because of low particle numbers in those bins?

The phrase is confusing and therefore reworded ad follows: The four variables (9.7 nm, 11.5 nm, 557.2 nm and 733.6 nm; midpoint of the merged three consecutive size bins) were set as weak along with the total variables (N10 – 800).

The four variables were downweighted due to high scale residuals in the preliminary runs. The total variable was set weak as explained variable.

Fig A10:

The lines represent the percentages, not a fitted lognormal distribution to the total PNSD. Wouldn't a meaningful factor look like a lognormal distribution? I think that the idea of PMF is that you add up the individual contributions (of different particle sources) to the total observed particle population. Similar to in AMS PMF, where the individual factors have to correspond to existing mass spectra (of reference

particles or other observed particle types). So here I would assume that the individual factors should look like size distributions as well. Some of the factors do not look like "typical" size distributions. But it may well be that I am wrong, so please explain and comment.

The figure was redone based on the PMF results including number size distribution (dN/dlogDp) as well as volume size distribution (dV/dlog Dp) of the factor profiles (Fig. A12).

Line 463 ff:

Here it becomes clear that the "high particle number concentration episodes" are not the same as S1-10 and W1-13. So W1-W8 from Table 4 are not the same as W1-W8 from Table A1? This is indeed confusing to the reader.

Since the whole manuscript was rearranged I hope that this will be no more an issue. We focused primarily on the episodes of high mass concentrations and the episodes of high number concentrations are presented in separated were put in the end of Section 3.6.

Line 481 ff:

The "episodes of high mass concentration" are the same as those from Table A1, as I guess now.

Yes, the episodes S1 – 10 and W1 – W13 presented in the Section 3.5 are the same as the one presented in the Section 3.4 and in the Table 1. The table reference was add to the sentence.

Table 4 vs Table 5:

Why did you use only one method for the effective density in Table 4 but both methods for Table 5?

The material density as well as Jayne shape factor and dynamic shape factor were calculated for eight winter episodes (N_W1 – N_W8) of high particle concentrations, the values were added to the Table 4 and discussed in the Section 3.5.

Fig A5 and A13:

I suggest moving Fig A13 and Fig A5 to the main text. There too many important figures in the Appendix.

Thank you for this suggestion. We have moved Figures A5 and A13 to the main text as Figs. 3 and 7, respectively. The other Figures have been renumbered.

Fig A13:

How does the plot look when you include the densities from Equ (1)?

This approach requires longer time span for the good spectra fit as we see by the longer episodes of high mass concentrations in comparison to the shorter episodes of high number concentrations where we calculated one density value for the whole episod. Therefore, even the hourly data for the diurnal trends calculation is not suitable due to low number of data points resulting in a higher uncertainty by spectra fits thus the density estimations.

Fig A14:

Move also Fig A14 to main text. Rather shift tables into the appendix (the reader likes to see figures not tables). Which equation was used? (it is missing in the caption). Furthermore, it seems that the fit to nitrate for summer do not really match the data points

Extrapolating the organic fraction would reach from rho = 1.2. at 100% org to rho = 1.77 for 0% organics. That fits very well to the "real" density. I therefore suggest to expand the y-axis and extend the linear fits.

Thank you for the suggestion. We have added original Figure A14 to the main text as Fig. 8. We used equation 2, which we added to the figure caption. The linear fit of nitrate is real, we verified by two programs. The visual deviation is probably caused by a large variance and also by a large number of values close to zero concentrations. The low correlation for nitrate corresponds to this.

We have added an ideal extrapolation of organics density to the summer Fig. 8. Compared to the "real" density there is some difference, which seems to indicate that the density for 0 % organics (i.e. BC) should be somewhere around 1.7 g cm$^{-3}$.

Regarding the density discussion, we have added new Figure A11 to the Appendix and following text to the Section 3.5: To be able to compare our results with above mentioned studies, we also used density of 1.2 g cm-3 for organics in Eq. (2). Therefore, as the mass fraction of organics in the aerosols increased, the density calculated using Eq. (2) converged to a value of 1.2 g cm-3 (Fig. 8). The use of higher density value for Org in Eq. (2) (e.g., 1.3 and 1.4 g cm-3) affects the overall density value, thus $\rho$m is more in agreement with $\rho$eff. Increasing value of the Org density in Eq. (2) also flatten the diurnal trend in winter, but it still holds significant diurnal variations (Fig. A11).

[Figure]

Change in legend description: "Fig. 8. Relationship between density, calculated according to Eq. 2, and mass fractions of the main NR-PM 1 species. Idealized extrapolation of organics densities is added to summer figure for $\rho$ = 1.2 g cm$^{-3}$ at 100% Org, and $\rho$ = 1.77 g cm$^{-3}$ for 0% organics."

[Figure]

Fig. A11. Diurnal trends of average ρm calculated based on Eq. (2) in winter for different organic densities (1.2, 1.3 and 1.4 g cm-3) in absolute (left) and normalized (right) values.

Lines 512 – 518:

The differences between Eq. 1 and 2 for the density is basically the shape factor (see DeCarlo).

Therefore, from comparison between (1) and (2) you can infer the shape of the particles. I think that's important piece of information you can retrieve here.

In the Section 2.2.2 the densities, effective density (ρeff) and material density (ρm), were defined as well as the Jayne Shape factor (S) and the dynamic shape factor (χ). In the Section 3.5 and 3.6 the results for the densities and shape factors are discussed.

 Summary and Conclusions:

This is only a summary. You repeat the findings that you listed in the individual sections of the paper. But are there any conclusions? I think that **the density results in Figs A13 and A14** are interesting enough to discus them further. As I said, Equ (2) yields the "real" density (rho_m in DeCarlo), while Equ (1) (which is Equ 44 in DeCarlo 2004) yields the effective density. If you include Equ [45] from deCarlo, you can infer the shape factor from the measured parameters.

 The Section 4. Summary and conclusions was rewritten.

**Technical comments**

Line 103: "1" is missing in "16.7 l min-1"

corrected

Line 105: "0.1-l min" -> "0.1 l min-1"

corrected

Line 105 – 108: Only the IE calibration is performed in BFSP mode, the other two don't require BFSP. Performing an IE calibration only at the beginning of the campaign is not best practice. Typically it's recommended      at      minimum      once      per      week      (http://cires1.colorado.edu/jimenezgroup/wiki/index.php/Field_ToF-AMS_Operation#Standard_Field_Calibration_procedures), but at least a second IE calibration should have been performed after the campaign.

The AMS size and flow as well as ionization efficiency (IE) calibrations in the brute-force single-particle mode (BFSP, Drewnick et al., 2005, monodisperse 350-nm ammonium nitrate aerosol particles) were performed in the beginning, during and after each campaign. The resulting IE was the average IE from all calibrations.

Fig A2: It is easier to compare if %-scales are the same

The figure was redone.

Line 242: Ammonium (ammonia is the gas NH3)

corrected

Fig 1: Right plot axis labels: µm -> µg

The figure with the wrong axis label presenting mass closure was deleted.

Line 148: The data are recorded a function of particle time-of-flight (PToF). 10 nm – 7000 nm is just an extrapolation of the size calibration curve to the PToF. It is not the actual size range (as you mention), because the aerodynamic lens does not transmit particles over this size range. But please delete this size range here because it is not real. A related question: which type of aerodynamic lens was used in this C-ToF-AMS?

In this C-ToF-AMS we use standard aerodynamic lens for transmitting particles below PM1.

Original text: "… AMS data as mass size distribution based on the vacuum aerodynamic diameter ($D_{va}$) in the size range from 10 to 7000 nm (calculated in Squirrelu software, 50 – 800 nm in reality)…" was changed to "… AMS data as mass size distribution based on the vacuum aerodynamic diameter ($D_{va}$) in the size range approximately from 50 to 800 nm (in Squirrel software extrapolated in range 10 – 7000 nm)"

Line 148: correct "Squirrelu"

corrected

Line 150: "he" -> "the"

corrected

**Our references:**

Beddows, D.C.S., Harrison, R.M., Green, D.C, Fuller, G.W., 2015. Receptor modelling of both particle composition and size distribution from a background site in London, UK. Atmos. Chem. and Phys. 15, 10107-10125.

Kubelová, L., Vodička, P., Schwarz, J., Cusack, M., Makeš, O., Ondráček, J., Ždímal, V., 2015. A study of summer and winter highly time-resolved submicron aerosol composition measured at a suburban site in Prague. Atmos. Environ. 118, 45–57.

Leoni, C., Pokorná, P., Hovorka, J., Masiol, M., Topinka, J., Zhao, Y., Křůmal, K., Cliff, S., Mikuška, P., Hopke, P.K., 2018. Source apportionment of number size distributions and mass chemical composition in a European air pollution hot spot. Environmental Pollution 234, 145-154.

Masiol, M., Vu, T. V., Beddows D. C. S., Harrison R. M., 2016. Source apportionment of wide range particle size spectra and black carbon collected at the airport of Venice (Italy). Atmos. Environ. 139, 56-74.

Pokorná, P., Leoni, C., Schwarz, J., Ondráček, J., Ondráčková, L., Vodička, P., Zíková, N., Moravec, P., Bendl, J., Klán, M., Hovorka, J., Zhao, Y., Cliff, S.S., Ždímal, V., Hopke, P.K., 2020. Spatial-temporal variability of aerosol sources based on chemical composition and particle number size distributions in an urban settlement influenced by metallurgical industry. Environmental Science and Pollution Research 27, 38631–38643.

Sowlat, M H., Hasheminassab S., Sioutas C., 2016. Source apportionment of ambient particle number concentrations in central Los Angeles using positive matrix factorization (PMF). Atmos. Chem. Phys 16, 4849-4866.

Zíková, N., Pokorná, P., Makeš, O., Sedlák, P., Pešice, P., Ždímal, V., 2020. Activation of atmospheric aerosol in fog and low clouds. Atmospheric Environment 230, 117490, 1–11.

**Reviewer's References**

Crippa, M., DeCarlo, P. F., Slowik, J. G., Mohr, C., Heringa, M. F., Chirico, R., Poulain, L., Freutel, F., Sciare, J., Cozic, J., Di Marco, C. F., Elsasser, M., Nicolas, J. B., Marchand, N., Abidi, E., Wiedensohler, A., Drewnick, F., Schneider, J., Borrmann, S., Nemitz, E., Zimmermann, R., Jaffrezo, J.-L., Prévôt, A. S. H., and Baltensperger, U.: Wintertime aerosol chemical composition and source apportionment of the organic fraction in the metropolitan area of Paris, Atmos. Chem. Phys., 13, 961-981, doi:10.5194/acp-13-961-2013, 2013.

Cubison, M. J., Ortega, A. M., Hayes, P. L., Farmer, D. K., Day, D., Lechner, M. J., Brune, W. H., Apel, E., Diskin, G. S., Fisher, J. A., Fuelberg, H. E., Hecobian, A., Knapp, D. J., Mikoviny, T., Riemer, D., Sachse, G. W., Sessions, W., Weber, R. J., Weinheimer, A. J., Wisthaler, A., and Jimenez, J. L.: Effects of aging on organic aerosol from open biomass burning smoke in aircraft and laboratory studies, Atmos. Chem. Phys., 11, 12049–12064, https://doi.org/10.5194/acp-11-12049-2011, 2011.

Hennigan, C. J., Miracolo, M. A., Engelhart, G. J., May, A. A., Presto, A. A., Lee, T., Sullivan, A. P., McMeeking, G. R., Coe, H., Wold, C. E., Hao, W.-M., Gilman, J. B., Kuster, W. C., de Gouw, J., Schichtel, B. A., Collett Jr., J. L., Kreidenweis, S. M., and Robinson, A. L.: Chemical and physical transformations of organic aerosol from the photo-oxidation of open biomass burning emissions in an environmental chamber, Atmos. Chem. Phys., 11, 7669–7686, https://doi.org/10.5194/acp-11-7669-2011, 2011.

Jimenez, J.L., M.R. Canagaratna, N.M. Donahue, A.S.H. Prevot, Q. Zhang, J.H. Kroll, P.F. DeCarlo, J.D. Allan, H. Coe, N.L. Ng, A.C. Aiken, K.D. Docherty, I.M. Ulbrich, A.P. Grieshop, A.L. Robinson, J. Duplissy, J. D. Smith, K.R. Wilson, V.A. Lanz, C. Hueglin, Y.L. Sun, J. Tian, A. Laaksonen, T. Raatikainen, J. Rautiainen, P. Vaattovaara, M. Ehn, M. Kulmala, J.M. Tomlinson, D.R. Collins, M.J. Cubison , E.J. Dunlea, J.A. Huffman, T.B. Onasch, M.R. Alfarra, P.I. Williams, K. Bower, Y. Kondo, J. Schneider, F. Drewnick, S. Borrmann, S. Weimer, K. Demerjian, D. Salcedo, L. Cottrell, R. Griffin, A. Takami, T. Miyoshi, S. Hatakeyama, A. Shimono, J.Y Sun, Y.M. Zhang, K. Dzepina, J.R. Kimmel, D. Sueper, J.T. Jayne, S.C. Herndon, A.M. Trimborn, L.R. Williams, E.C. Wood, C.E. Kolb, A.M. Middlebrook, U. Baltensperger, and D.R. Worsnop, Evolution of Organic Aerosols in the Atmosphere, Science, 326, 1525-1529, doi: 10.1126/science.1180353, 2009.

Milic, A., Mallet, M. D., Cravigan, L. T., Alroe, J., Ristovski, Z. D., Selleck, P., Lawson, S. J., Ward, J., Desservettaz, M. J., Paton-Walsh, C., Williams, L. R., Keywood, M. D., and Miljevic, B.:

Biomass burning and biogenic aerosols in northern Australia during the SAFIRED campaign, Atmos. Chem. Phys., 17, 3945–3961, https://doi.org/10.5194/acp-17-3945-2017, 2017.

Ng, N. L., Canagaratna, M. R., Zhang, Q., Jimenez, J. L., Tian, J., Ulbrich, I. M., Kroll, J. H., Docherty, K. S., Chhabra, P. S., Bahreini, R., Murphy, S. M., Seinfeld, J. H., Hildebrandt, L., Donahue, N. M., DeCarlo, P. F., Lanz, V. A., Prévôt, A. S. H., Dinar, E., Rudich, Y., and Worsnop, D. R.: Organic aerosol components observed in Northern Hemispheric datasets from Aerosol Mass Spectrometry, Atmos. Chem. Phys., 10, 4625–4641, https://doi.org/10.5194/acp-10-4625-2010, 2010.

---

## Author Comment (AC2)

We would like to thank the reviewer for the comments that helped to improve the present study. Responses to each comment are provided below - see blue text.

**REVIEW #2**

The authors report data from a summer and a winter campaign at a monitoring site in Central Europe. The data are valuable and seem to meet the required quality standard. However, the presented analysis and interpretation is not sufficient and does not meet the standard of a publication in ACP. Many things are mentioned and connections are suggested without supporting arguments. There is no focused line that guides the reader from the measurements to the lessons that can be learned. What are the new insights and which results of the study confirm observations from previous studies? Unfortunately, I found no answers to these question, but reading this manuscript left me behind in confusion and I still wonder: What exactly can be learned from this study? Therefore I cannot recommend the publication. But I encourage the authors to improve their analysis and I hope that they find the comments below helpful.

Major comments:

The PMF analysis has been used to identify episodes of high number concentration. However it remains obscure how exactly this has been done. More explanation is needed (maybe in the appendix). Therefore, it remains unclear on which criteria the 10 summer and 13 winter episodes have been selected. In addition, these periods should be marked in Figure 2 (not in the appendix)!

PMF of PNSD is described in Section A1 (in Appendix) along with two new figures A12 and A13.

The criteria for the episode selection is described in the Section 2.2.4: … episodes of high mass concentrations were chosen based on a set of criteria: high mass size distribution of at least one main NR-PM1 species corresponding to the season ($NO_3^-$ ≥ 0.5/0.2 μg m⁻³, $SO_4^{2-}$ ≥ 1/0.5 μg m⁻³, $Org$ ≥ 6 /2 μg m⁻³); monomodal mass size distribution of all main NR-PM1 species; duration of the episodes min 1.5 hour. Also, the episodes were marked as well in Figure 2.

The discussion of the episodes is not convincing and too strong conclusions are drawn from the trajectory calculations without providing additional evidence. In lines 314/315 the authors link high nitrate concentrations to marine air masses. What are the arguments here? Do the authors suggest a marine source of nitrate? I do not see evidence for such a claim. In lines 328/329 maritime influence is suggested again. Given that the air crossed half a continent at low altitudes before being sampled at the site, the claim of marine influence is not convincing.

The discussion in lines 327-334 is confusing. Too many different things are mentioned nothing is followed up and or backed by arguments. For example, what is the evidence for inversion conditions in Central Europe during episode 6a? Why would this lead to higher NO3- without increasing organics as well? Moreover, the mixing layer height in Fig A3 suggests stronger inversion during W6b. What do the aauthors concude from f60 and Figure A8 that is mentioned but no interpretation is provided.

The episodes of high nitrate concentrations (lines 314/315) were discussed as follows: The episodes of high $NO_3^-$ concentrations were mainly linked to continental air masses (from the NW-SW, Fig. A6) from northern France, Benelux, central Germany and northern Italy. These regions were traced as hotspots of high particulate nitrate concentrations related to intense agricultural activities under anticyclonic conditions in late-winter and early-spring (Waked et al., 2014; Petit et al., 2017, 2019; Favez et al., 2021).

Line 328/329 was reworded as follows: The highest concentrations of Org (15.63 μg m$^{-3}$) as well as low concentrations of $SO_4^{2-}$, $NO_3^-$ and $NH_4^+$ (0.74 μg m$^{-3}$, 0.93 μg m$^{-3}$ and 0.96 μg m$^{-3}$, respectively) measured in winter during W7 were influenced by maritime air masses crossing France and Germany before reaching the NAOK (Fig. A6).

The evidence of the inversion conditions in Central Europe during episode W6a is the mid-boundary layer height calculated using HYSPLIT for several sites (Leipzig, Munich, Vienna, Katowice and Wroclaw) see the figures below. The mid-boundary layer height ranged from 25 m AGL (Munich) to 91 m AGL (Vienna). At the NAOK the mid-boundary layer height was during for W6a 42 m AGL and W6b 155 m AGL see the figures below. Therefore, we suggested stronger inversion during W6a episode.

[Figure]

[Figure]

[Figure]

[Figure]

High concentrations of $NO_3^-$ during W6a are discussed as follows: … the highest $NO_3^-$ concentrations (10.66 μg m$^{-3}$) measured in the W6a episode were characterized by below-freezing temperatures, which probably arose due to inversion conditions in Central Europe. The conditions prevailing during the W6a episode, in combination with ammonia due to the agricultural activities, including the spreading of fertilizers, induce increase of particulate nitrate and ammonium concentrations similarly as reported Favez et al., 2021 for Northern France.

Fig. A7 compares organic fragments f44 and f60, which enables us to assess the presence of fresh or aged organic aerosols emitted by BB during the monitored periods.

Lines 335-337: Why does the **oxidation state** in winter point out the importance of **local sources**? Hydrocarbon aerosol can be transported as well and will remain hydrocarbon in the absence of significant photochemistry.

New figure showing origin of organic aerosol at Košetice site in winter was added and the text was rephrased as follows:

Organic aerosol ageing was examined on the f44 and f43 fragments (Fig. 3). Winter aerosols were **less oxidized** than summer aerosols. This results along with the organics diurnal trends of late evening maxima (Fig. 3) and polar plots (Fig. A5) pointing to the importance of local combustion sources during the cold  part of the year. Importance of local fosile fuels combustion for residential heating as a source of fresh OA/ hydrogen-like OA in winter is presented in study by Chen et al., 2021 (under review).

Chen et al., 2021. European Aerosol Phenomenology - 8: Harmonised Source Apportionment of Organic Aerosol using 22 Yearlong ACSM/AMS Datasets. Environmental International (under review).

[Figure]

Fig. A5. Polar plots showing the origin of organics in summer (left) and winter (right).

Lines 338-339: How were MOOAs and LV-OOA retrieved? There is no information on this in the manuscript.

The sentence was left out.

Lines 347-353: This may be interesting. I understand that levoglucosan has been measured (although this is not -but should be - mentioned in the method section) about 10 times higher in the winter samples than during summer, but the f60 parameter barely indicates biomass burning influence. In my opinion this could be explored more. Apparently there is a discrepancy and something can be learned here!

In the method section 2.1 PM1 filter analysis by IC for monosaccharide anhydrides is mentioned, however levoglucosan was specifically added to the text as follows:

 Additionally, 12-h PM1 filter samples were collected by a sequential Leckel LVS-3 (Sven Leckel Ingenieurbüro, Germany) for subsequent chemical analyses of cations, anions and monosaccharide anhydrides (levoglucosan, mannosan and galactosan)  using ion chromatography (Dionex ICS-5000+ system, Sunnyvale, CA, USA).

lines 353-358: I do not understand this discussion and I do not agree that such conclusions can be drawn from Figure A9.

The sentence was rephrased and the number of the figure was corrected as follows: Additionally, a comparison of fragments f44 and f60 enabled us to assess the presence of fresh or aged organic aerosols emitted by BB (e.g., Milic et al., 2017) revealing that aged organic aerosols from BB influenced the site during both seasons especially in winter (Fig. A7).

The discussion in lines 359-377 is not convincing. Are the differences observed between the clusters statistically significant at all? In most cases it seems that they are not.

Yes, there were statistical differences among clusters (summer – 6 clusters, winter – 5 clusters) by all variables using Shapiro-Wilk normality test and Kruskal-Wallis rank sum test at the alpha value 0.05. The results are discussed in the text.

Section 3.4 (lines 388-442) could use more focus and I wonder if all the evidence from other studies for larger particles in winter is needed. More space should be dedicated instead to interpret the findings of this study. The potentially interesting Figure 5 is discussed in less than 3 lines! Certainly much more can be inferred here.

We would like to thank reviewer for his opinion. The whole paragraph was added discussing results of the figure presenting relationship between organic fragment f44 and the size of the organic fraction during episodes of high NR-PM1 species mass concentrations in both seasons.

In section 3.5 the authors focus on density retrievals during the defined episodes. The purpose of this is not clear. It is stated that a density of 1.85 g/cm3 corresponds to black carbon, but certainly nobody is claiming that these particles were predominantly black carbon. So what can be learned from this analysis? Moreover, there is no discussion of uncertainties of this analysis. is a value of 1.45 different than a value of 1.55? or is this still in the range of uncertainty?

The Section was rewritten with a focus on the episodes of high mass concentrations. In this section we discussed results of particle densities (effective and material) and shape factors (Jayne shape factor and dynamic shape factor) calculations as well as the seasonal differences between these parameter.

The density uncertainty from the first approach (density estimate from mobility and aerodynamic measurements, Eq. 1) was estimated to be +- the smallest increment used in the analysis, i.e. 0.05 g cm$^{-3}$. The difference of 0.05 g cm$^{-3}$ is well visible in the results (taking the uncertainty of the sizing of SMPS – within 3%, see Wiedensohler et al. 2017 and AMS – within 8%, see Takegawa et al., 2005); for example, below the resulting density of 1.55 in winter episode #7 was plotted together with 1.50 and 1.60 (light blue dashed lines). The density uncertainty is mentioned in the Section 2.2.2 Particle density and shape factor estimation.

[Figure]

Takegawa, N., Miyazaki, Y., Kondo, Y., Komazaki, Y., Miyakawa, T., Jimenez, J.L., Jayne, J.T., Worsnop, D.R., Allan, J.D., Weber, R.J., 2005. Characterization of an Aerodyne Aerosol Mass Spectrometer (AMS): Intercomparison with Other Aerosol Instruments, Aerosol Science and Technology, 39:8, 760-770.

Wiedensohler, A., Wiesner, A., Weinhold, K., Birmili, W., Hermann, H., Merkel, M., Müller, T., Pfeifer, S., Schmidt, A., Tuch, T., Velarde, F., Quincey, P., Seeger, S., Nowak, A., 2017. Mobility particle size spectrometers: Calibration procedures and measurement uncertainties. Aerosol Science and Technology, 52:2, 146-164.

Minor comments:

Abstract: abbreviations 'SE' and 'SW' not explained. Better to spell this out...

The abbreviations were spelled out.

The abstract is a list of observations. But what are the lessons learned from the study? What can we conclude from the observations.

The abstract was partially rewritten.

Showing the two almost identical charts in Figure 1 is not necessary in my opinion. In theory the volume based approach should be a little bit more advanced because it takes the particle composition into account to some extent, while the other approach uses a constant density for all particles and seasons.

Yes, we agree, therefore only the result of the volume based approach is presented.

Fig A5: add a legend. The reference "common color code" is not sufficient.

The legend was added.

Fig A14: incomplete caption. ???

The figures were redone and removed to the main text as Figure 8.

Fig A10: remaiuns unclear in the current form. More explanation is needed.

The figure was redone based on the PMF results including number size distribution (dN/dlogDp) as well as volume size distribution (dV/dlog Dp) of the factor profiles (Fig. A12).

---

## Author Response (AR2)

We would like to thank the reviewer for the positive feedback to the paper revision. Responses to each comment are provided below - see blue text.

Dear authors,

the paper has been revised and considerably improved. Thank you very much for answering my questions and revising the manuscript accordingly. I suggest to accept the manuscript with some minor corrections that I list here:

Line 157/158: Squirrel does this extrapolation, but it should no be used. I would not mention it here.

The text in the bracket was deleted.

Section 2.2.3 I suggest "trajectory analysis" as section heading

The section was renamed according the reviewer's suggestion.

Fig 8 and lines 503-517: Why don't you show the idealized extrapolation of the organic density also for winter? Would it fit? I think we can learn something here, also with respect to Fig A11.

The idealized extrapolation was added to the winter Fig 8. The fit is similar as the one in summer. The scattered values of all species in winter in comparison to summer are probably due to periods of low concentrations depicted in Fig. A3 and reflected in the Fig A11.

line 546: was -> were

The error was corrected.